# Spatial–temporal order–disorder transition in angiogenic NOTCH signaling controls cell fate specification

Tae-Yun Kang[1,2†], Federico Bocci[3,4†], Qing Nie[3,4], José N Onuchic[5]*, Andre Levchenko[1,2]*

[1]Department of Biomedical Engineering, Yale University, New Haven, United States; [2]Yale University, New Haven, United States; [3]NSF-Simons Center for Multiscale Cell Fate Research, University of California Irvine, Irvine, United States; [4]Department of Mathematics, University of California Irvine, Irvine, United States; [5]Center for Theoretical Biological Physics, Rice University, Houston, United States

*For correspondence:
jonuchic@rice.edu (JNO);
andre.levchenko@yale.edu (AL)

†These authors contributed equally to this work

Competing interest: The authors declare that no competing interests exist.

**Abstract** Angiogenesis is a morphogenic process resulting in the formation of new blood vessels from pre-existing ones, usually in hypoxic micro-environments. The initial steps of angiogenesis depend on robust differentiation of oligopotent endothelial cells into the Tip and Stalk phenotypic cell fates, controlled by NOTCH-dependent cell–cell communication. The dynamics of spatial patterning of this cell fate specification are only partially understood. Here, by combining a controlled experimental angiogenesis model with mathematical and computational analyses, we find that the regular spatial Tip–Stalk cell patterning can undergo an order–disorder transition at a relatively high input level of a pro-angiogenic factor VEGF. The resulting differentiation is robust but temporally unstable for most cells, with only a subset of presumptive Tip cells leading sprout extensions. We further find that sprouts form in a manner maximizing their mutual distance, consistent with a Turing-like model that may depend on local enrichment and depletion of fibronectin. Together, our data suggest that NOTCH signaling mediates a robust way of cell differentiation enabling but not instructing subsequent steps in angiogenic morphogenesis, which may require additional cues and self-organization mechanisms. This analysis can assist in further understanding of cell plasticity underlying angiogenesis and other complex morphogenic processes.

## eLife assessment

The authors used an appropriate micro-engineered experimental model of angiogenesis coupled to mathematical model to study the early steps of the angiogenic sprouting. To this end, the authors developed a **convincing** model to predict how VEGF activates Delta-Notch signaling. The work affords **important** new insight into the complex processes involved in the onset of angiogenesis.

## Introduction

Angiogenesis, that is, the formation of new blood vessels from the pre-existing ones, is a striking example of phenotypic plasticity in an adult differentiated endothelium. Pro-angiogenic factors secreted in response to hypoxic conditions, particularly the vascular endothelial growth factor (VEGF), specify differentiation of endothelial cells lining blood vessels into diverse phenotypic states, including the pro-migratory Tip cell phenotype. Tip cells can initiate invasive cell migration into the surrounding extracellular matrix (ECM), leading to sprouting and branching of the nascent vessels (*Adams and Alitalo, 2007*; *Potente et al., 2011*). Tip cells are differentiated

**eLife digest** Blood vessels are vital for transporting blood containing oxygen, nutrients and waste around the body. To maintain this function, new blood vessels are continually formed through a process called angiogenesis. Often triggered in areas requiring oxygen, new blood vessels form from existing vessels as 'sprouts' in response to elevated levels of a signaling molecule called vascular endothelial growth factor (or VEGF for short).

For 'sprouting' to occur, endothelial cells lining the parental blood vessel must become either 'Tip' or 'Stalk' cells. Tip cells lead the extension of the blood vessel sprouts, while Stalk cells proliferate rapidly, ensuring the growth of the sprout. Correct spatial arrangement of these different cell types is crucial for the development of functional blood vessels.

Previous work has shown that VEGF promotes differentiation of endothelial cells lining blood vessels into different cell types. In neighboring cells, a signaling pathway known as NOTCH is activated due to interactions between adjacent cells, promoting differentiation of Tip cells and Stalk cells. Ideally, Tip cells are spaced out by intervals of Stalk cells to allow separate sprouts to form. Throughout this process, a single cell can receive contradictory signals, with VEGF promoting Tip cell formation and NOTCH signaling promoting Stalk cell differentiation. It remained unclear how the right cells are formed in the right places when surrounded by these conflicting inputs.

To better understand these dynamics Kang, Bocci et al. combined a laboratory model of angiogenesis with mathematical modelling. Experiments using these approaches showed that the overall pattern of cell type specification induced by VEGF and NOTCH signaling is consistent with so-called order-disorder transition, commonly observed in crystals in other ordered structures. For blood vessel cells, this transition means that they can still robustly take on either the Tip or Stalk cell identities, but this fate selection is not stable in time. Additionally, the overall pattern is much more sensitive to additional cues and self-organization mechanisms. Further analysis revealed that one such cue can be local fluctuations the density of fibronectin, a key pro-angiogenic extracellular component, leading to formation of sprouts that tend to distance themselves as much as possible from other fully formed sprouts.

These findings provide a framework for understanding NOTCH-mediated patterning processes in the context of responding to a variety of environmental cues. This sensitivity in cell type specification is important for determining the dynamic nature of the initial steps of angiogenesis and may be crucial for understanding growth of new blood vessels in damaged organs, cancer and other diseases.

from Stalk cells, another phenotypic state, through juxtacrine cell–cell interaction between these cell types involving NOTCH1 signaling, triggered and modulated by induction of Dll4 and Jag1 ligands (*Benedito et al., 2009*; *Blanco and Gerhardt, 2013*). Stalk cells can therefore form in immediate proximity of Tip cells, particularly, at the leading edge of an extending sprout, if the NOTCH signaling is sufficiently pronounced for the Tip–Stalk differentiation to occur. Proliferation of Stalk cells is as essential as the invasive migration of Tip cells for the emergence, extension and branching of growing sprouts, making the analysis of coordinated Tip and Stalk specification particularly important.

The inputs specifying the cell fate can be potentially contradictory, for example, with pro-angiogenic factors, such as VEGF, promoting the Tip cell fate, and the NOTCH signaling activated by the neighboring cells promoting the Stalk cell fate and thus suppressing the Tip cell identity in the same cell. These and other signaling inputs can thus be incoherent in terms of cell fate specification and can result in complex dynamic outcomes that are still poorly understood. Further elucidation of these processes thus requires high-resolution, quantitative experimental measurements tightly coupled with computational analysis. Since such measurements are still challenging in vivo, particularly in mammalian tissues, use of tissue models recapitulating the salient features of the developing vasculature is a key tool in the current analysis of angiogenesis and development of de novo vascular beds.

Previously, we and others have developed a set of micro-fabricated experimental angiogenesis models that have had progressively improved biomimetic characteristics (*Wang et al., 2020*; *Liu et al., 2021*; *Kang et al., 2019*; *Chen et al., 2017*; *Nguyen et al., 2013*). These characteristics include spatially and biochemically appropriate cell micro-environments, composed of components of the

ECM and of gradients of growth factors and cytokines around the developing vasculature, which is composed of endothelial cells and pericytes.

We have previously used this approach to map different combinations of VEGF and an inflammatory cytokine, Tumor necrosis factor (TNF), onto pro- and anti-angiogenic outcomes, modeling frequently encountered angiogenesis conditions (*Kang et al., 2019*). This analysis provided evidence that 'mini-sprouts' — one-cell structures protruding from the parental blood vessel into the surrounding matrix — were comprised of Tip cells. However, it was not clear whether all such 'mini-sprouts' would ultimately develop into more mature multicellular sprouts with defined lumens and the potential to form new blood vessels. Furthermore, although our analysis was successful in explaining the fraction of Tip cells formed under different conditions, it was not clear how to account for the spatial aspects of the Tip cell and mature sprout specification, such as their mutual separation and density.

We address these challenges here by extending our analysis to a higher temporal and spatial resolution, both in experimental and mathematical models of angiogenic sprouting. Surprisingly, we found that the formation of mini-sprouts was a highly dynamic process, in which they could either retract after extension or form full-fledged sprouts. Furthermore, the experimentally determined spatial positioning of mini-sprouts was well explained by the predicted locations of the Tip cells in the mathematical model but the model could not account for which of the mini-sprouts would become fully formed sprouts. Further analysis revealed that the stable sprout formation from mini-sprouts can be enabled by the local fluctuations of the density of fibronectin, a key pro-angiogenic ECM component, leading to sparse patterns where sprouts tend to maximally distance themselves from other fully formed sprouts. These results reveal some of the key mechanisms that may define the density of the angiogenically formed vascular beds under diverse conditions.

## Results

### Dynamic angiogenesis can be explored in a 3D biomimetic experimental setup

To investigate the properties of angiogenic patterning and cell fate specification, we used an experimental model previously employed to assess the crosStalk between pro-angiogenic and pro-inflammatory stimuli (*Kang et al., 2019*). In this experimental setup, angiogenesis occurs from a 3D parental engineered endothelial vessel embedded in the collagen matrix and exposed to exogenously supplied VEGF and other pro-angiogenic factors (*Figure 1A, B*). In agreement with prior observations, we found that this setup resulted in formation of both one-cell extensions into the matrix (mini-sprouts) and full-fledged multicellular sprouts containing detectable lumens and pronounced leading Tip cells (*Figure 1C*). Sprouts displayed a variety of growth stages, including the very early ones, composed of one lumenized cell or pairs of connected cells, also forming a lumen (*Figure 1— figure supplement 1*). Although mini-sprouts formed throughout the observation area of the vessel, sprouts developed within specific zones, while other zones remained devoid of detectable sprout formation over the course of the study (*Figure 1D–F*). These observations suggested that cell fate specification and sprout formation are dynamic processes that may display diverse local outcomes. We therefore set out to characterize these processes in the context of an accessible and well-defined analysis tool that can allow to contrast experimental findings with mathematical models of angiogenic patterning, particularly those based on the commonly assumed NOTCH receptor-mediated cell–cell interactions (*Figure 1G*).

### Mathematical model of VEGF/NOTCH signaling predicts spatially resolved Tip–Stalk patterns

To set the framework for the analysis of cell fate determination, we extended our previously developed and experimentally validated mathematical model of Tip–Stalk fate differentiation between two cells (*Boareto et al., 2015b*) to a multicellular hexagonal lattice in two dimensions. In the new model, we replicated within each cell the signaling network incorporating the NOTCH and VEGF pathways (*Figure 2A*). Prior analysis of this model on the level of two adjacent cells predicted the emergence of bistability between a (high NOTCH, low Delta) Stalk phenotype and a (low NOTCH, high Delta) Tip phenotype (*Boareto et al., 2015b*). This result is consistent with the overall expectation of the differentiation effect of Delta–NOTCH signaling, which in 2D is further expected to

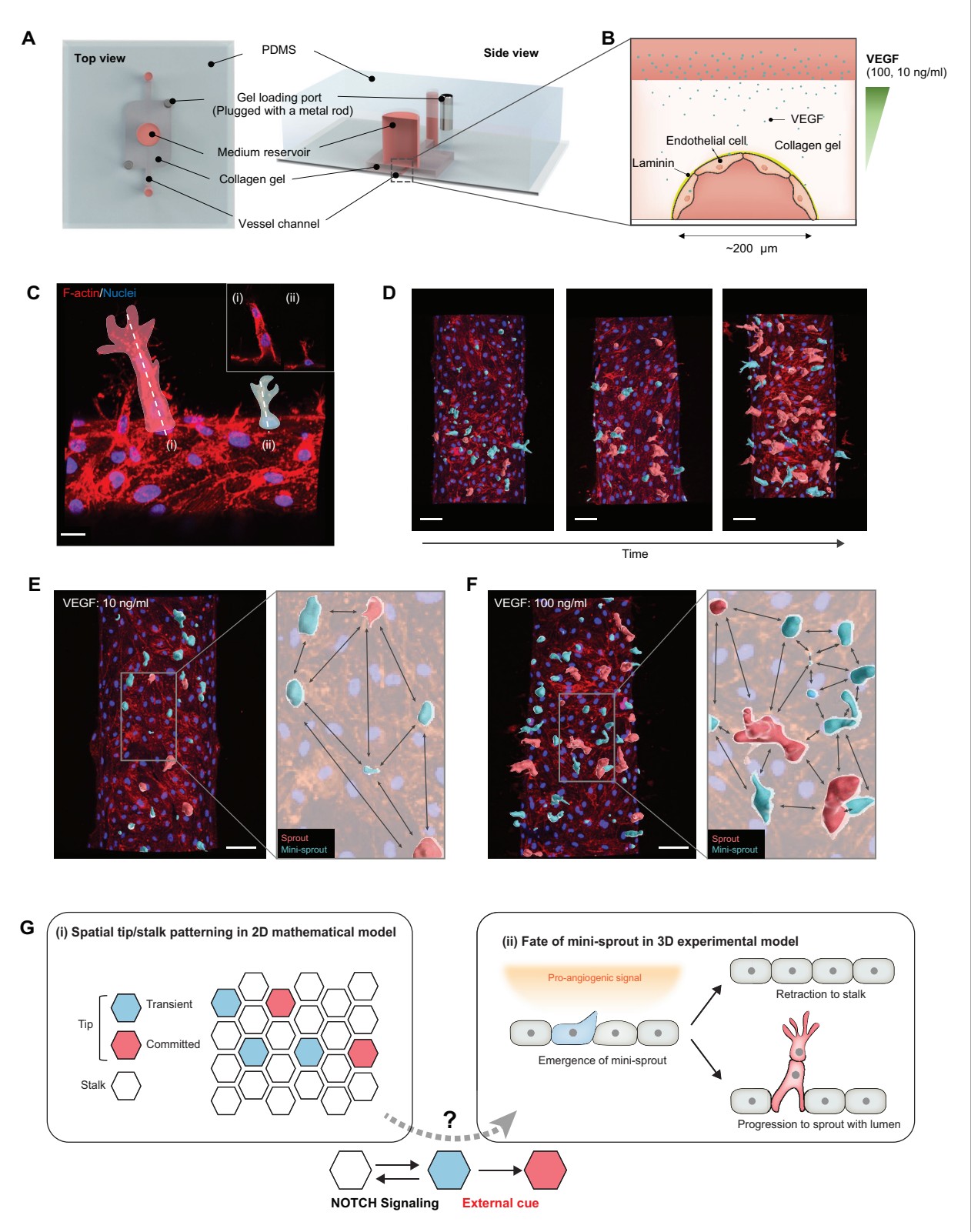

**Figure 1.** Analysis of temporal and spatial regulation of angiogenic fate specification in a 3D biomimetic experimental setup. (**A**) 3D vessel model for inducing angiogenesis in response to a gradient of vascular endothelial growth factor (VEGF). (**B**) Cross-section of a vessel embedded in collagen type I within the device; VEGF is added to the medium reservoir above the vessel to generate a VEGF gradient. (**C**) Angiogenesis leads to the formation of new sprout-related structures from the parental vessel that have two distinct morphologies: (i) full-fledged multicellular sprouts containing detectable

*Figure 1 continued on next page*

*Figure 1 continued*

lumens and (ii) mini-sprouts in the form of single cell extension into the matrix. Scale bars: 20 μm. (**D**) Temporally resolved observation of dynamic formation of sprouts and mini-sprouts populations during angiogenesis. As depicted in (**C**), sprouts are pseudo-labeled with red color and mini-sprouts in blue color. Scale bars: 50 μm. Dependence of the spatial distribution of sprouts and mini-sprouts on the VEGF concentration: (**E**) 10 ng/ml and (**F**) 100 ng/ml. Scale bars: 50 μm. Images are 3D reconstructions of confocal z-stacks, showing nuclear (Hoechst 33342) and cytoskeleton (Phalloidin). (**G**) Schematic overview of Tip–Stalk patterning: (i) Spatial Tip–Stalk patterning due to juxtacrine NOTCH signaling that might lead to fixed persistent and transient cell fate specification. (ii) Fates of mini-sprouts in experiments: both retraction (thus conversion from the phenotypically Tip to as Stalk phenotype) and stabilization and growth to a fully defined sprout are observed.

The online version of this article includes the following figure supplement(s) for figure 1:

**Figure supplement 1.** Phenotypic categorization of sprout and mini-sprout.

generate 'salt-and-pepper' patterns, with a single Tip cell surrounded by six Stalk cells (*Bocci et al., 2020*), yielding the overall fraction of Tip cells in this arrangement of 25%. However, this simple bistability and spatial patterning picture can be altered by signaling inputs that potentially conflict with those involved in Delta–NOTCH signaling (*Figure 1G*). For example, the VEGF pro-angiogenic factor promoting the Tip cell fate, can conflict with the NOTCH signaling activated by the neighboring cells that instead promotes the Stalk cell fate while suppressing the Tip cell identity. This might result in disordered patterns with adjacent Tip cells that deviate from the archetypical salt-and-pepper configuration. To explore the properties of this disordering effect, we ran simulations, in which the VEGF–NOTCH signaling occurred in all individual cells within a hexagonal array of the model multicellular endothelium, starting from randomized initial conditions (*Figure 2A*). The fully equilibrated patterns were then analyzed for distributions of the simulated Delta and NOTCH expression across the cells. We found for a wide range of VEGF inputs that the distributions of Delta and NOTCH displayed largely bimodal distributions, and the levels of the average Delta expression increased with the increasing input (*Figure 2B, C*, *Figure 2—figure supplement 1A, B*) due to positive effect of the activated VEGF receptor on Delta (see again the circuit in *Figure 2A*). Nevertheless, the clear overall bimodality allowed us to consistently classify cells into the Delta-high (Tip) and Delta-low (Stalk) cell states and examine the spatial distribution of these cellular subtypes (see Method: Definition of Tip cells in the model).

The spatial Tip–Stalk cell distribution patterns revealed a complex dependency on the VEGF input. At relatively low VEGF levels, the patterns were mostly ordered, with small deviations from the expected 'salt-and-paper' geometry with a 25–75% ratio of Tip–Stalk (*Figure 2D*). However, as the VEGF input increased, the fraction of Tips grew and the patterns became sharply more disordered over a relatively narrow range of magnitude of the VEGF input, which could be identified as a highly sensitive area separating more 'ordered-like' and 'disordered-like' patterns. Finally, increasing VEGF stimuli beyond the highly sensitive area further increased the disorder of the patterns, but with a lower VEGF sensitivity, over several more orders of magnitude of VEGF inputs (*Figure 2D, E* and *Figure 2—figure supplement 1A, B*). Spatial patterns in the disordered phase at high VEGF input levels were characterized by much higher fractions of Tip cells that were frequently in contact with each other as quantified by a 'disorder index' (*Figure 2—figure supplement 1C, D*). This transition was reminiscent of the order–disorder transition commonly observed and studied in the change of the order of atoms in various substances as a function of temperature, and in other chemical systems (*Yang et al., 2017*; *Bu et al., 2022*). As expected for these types of transitions, increasing or decreasing the order-stabilizing Delta–NOTCH cell–cell signaling, resulted in the corresponding shifts of the ranges of VEGF inputs, over which the sharp order–disorder transition occurred. For example, a small increment or decrease in the cellular production of the Delta ligand shifted the order–disorder transition to either lower or higher VEGF inputs (*Figure 2F*). As the broad sweep of the VEGF inputs simulated in the mathematical model is likely beyond the range of receptor sensitivity for the experimental VEGF signaling, we contrasted the predicted fractions of Tip cells with the previously made observations in the 3D experimental angiogenesis model shown in *Figure 1*. We found that, for the VEGF = 10 ng/ml, the experimentally determined fraction of the Tip cells was 0.02 ± 0.08 (mean ± standard deviation [SD]), that is, encompassing the fraction of 0.25 expected for the completely ordered, 'salt-and-pepper' Tip cell distribution pattern. However, for VEGF = 100 ng/ml the experimentally observed Tip cell fraction was 0.32 ± 0.01 (mean ± SD), which corresponded to the disordered state predicted by the model under higher VEGF inputs. We therefore concluded that the transition between order

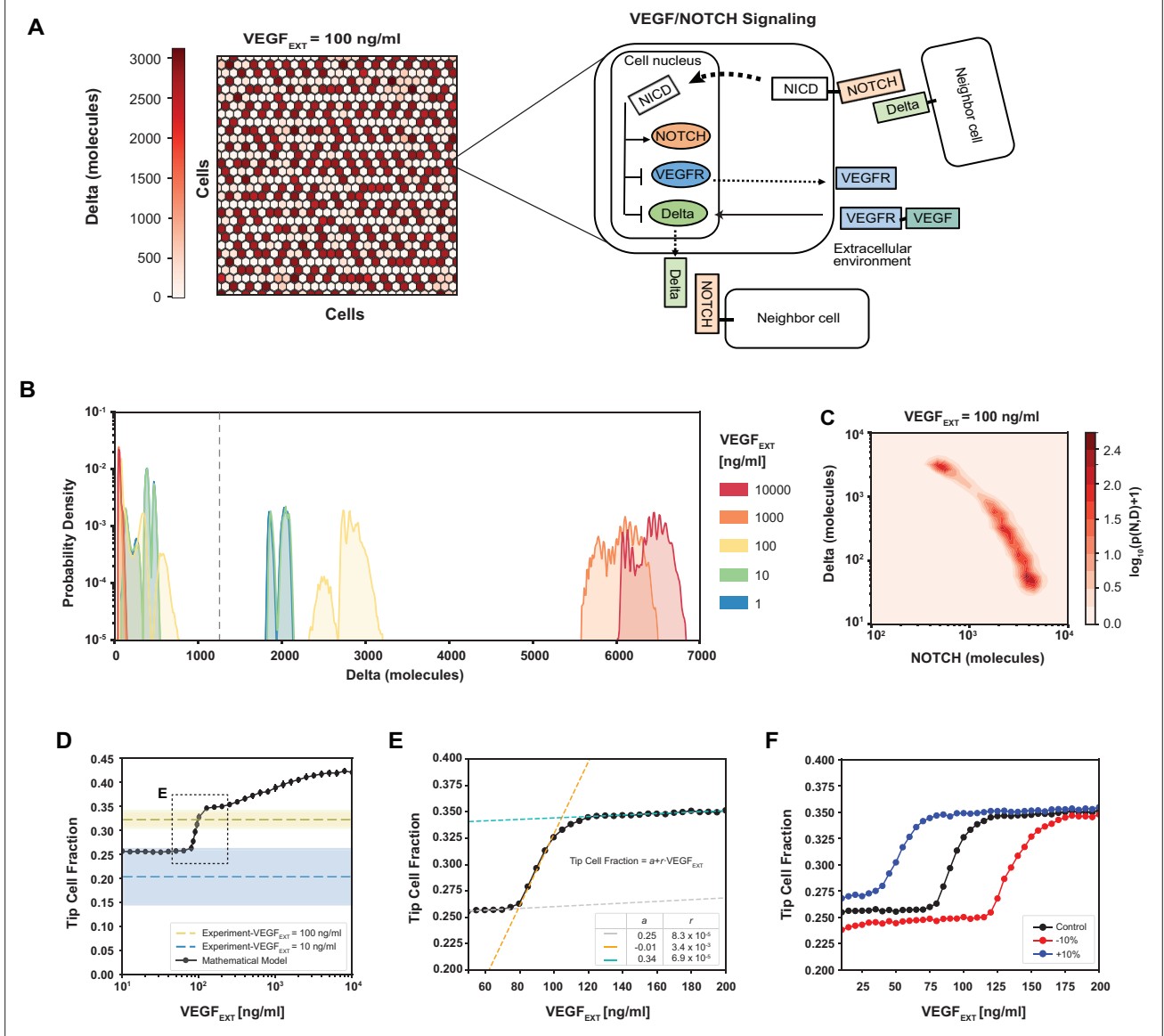

**Figure 2.** Robust differentiation and order–disorder transition are suggested by mathematical and experimental analyses. (**A**) Right: An example of a pattern after full equilibration on a 30 × 30 hexagonal lattice. Color scale highlights the intracellular levels of Delta. Left: The circuit schematic highlights the components of the intracellular NOTCH–vascular endothelial growth factor (VEGF) signaling network. (**B**) Distribution of intracellular Delta levels in the two-dimensional lattice for increasing levels of external VEGF stimuli. (**C**) Pseudopotential landscape showing the distribution of intracellular levels of NOTCH and Delta for VEGF$_{EXT}$ = 100 ng/ml. (**D**) Fraction of Tip cells as a function of external VEGF stimulus (black curve). Blue and yellow lines and shading depict experimental fractions of Tip cells for VEGF$_{EXT}$ = 10 ng/ml and VEGF$_{EXT}$ = 100 ng/ml, respectively. (**E**) Detail of Tip cell fraction transition zone (corresponding to box in panel D). Legend depicts the coefficients of the linear fits. (**F**) Shift of the VEGF$_{EXT}$ transition threshold upon variation of the NOTCH–Delta-binding rate constant. For panels B–F, results are averaged over 50 independent simulations starting from randomized initial conditions for each VEGF$_{EXT}$ level (see Methods: Simulation details).

The online version of this article includes the following figure supplement(s) for figure 2:

**Figure supplement 1.** Order–disorder transition in the NOTCH–Delta–vascular endothelial growth factor (VEGF) multicell model.

and disorder can occur in the 10–100 ng/ml range of VEGF concentration, allowing us to investigate the properties of the disordered state and its relationship to the spatial frequency of sprout formation. Based on the Tip/Stalk cell ratio, we calibrated the model's parameters so that a VEGF input of 100 ng/ml matched the experimentally observed Tip/Stalk fractions at the same experimental VEGF input (see Methods: Mathematical model of VEGF/NOTCH signaling).

## Precise quantification of Tip cell spatial arrangement suggests disordered patterning in the engineered angiogenesis model

To enable the comparison between the modeling predictions and experimental observations, we first quantified the spatial patterning characteristics of the inferred positions of the Tip cells in the experimental angiogenesis model. As in our prior analysis using this experimental approach (*Kang et al., 2019*), we identified Tip cells based on their key *phenotypic* characteristic — invasive migration into the surrounding collagen matrix. As suggested above, Tip cells can either be present in the form of 'mini-sprouts' or be at the Tips of sprouts containing recognizable lumens (see *Figure 3A* and *Figure 1—figure supplement 1* for examples of this classification). Since formation of both mini-sprouts and lumenized sprouts involved emergence of Tip cells specified in endothelial cell monolayer lining the parental vessel, we constructed a two-dimensional map of the experimentally inferred spatial positions of Tip cells at the location of all mini-sprouts and sprouts (*Figure 3A , B*). This mapping (see Methods: Quantification of Tip–Tip cell distance in experiments) assumed that the Stalk cells found in the extending sprouts emerge through cell proliferation, rather by Stalk cell migration from the parental vessel. Experimentally, we used nuclear staining to identify non-Tip cells. Since the results above suggested that the cell fates and their patterns in our experimental setup were consistent with the ordered or somewhat disordered 'salt-and-paper' patterns, we further assumed that all the non-Tip cells adopted the Stalk fates (in the sense of Delta-low, NOTCH-high status, alternative to the Tip cell fate). The resulting map of experimentally specified Tip and Stalk cell locations was then used to calculate the shortest distances between Tip cells, measured in 'cell hops', that is, the minimal number of intermediate cells between randomly chosen pairs of Tip cells (*Figure 3B*). These distances included Stalk cells exclusively, and no intermediate Tip cells. If two or more Tip cells were at equal distance from a given other Tip cell, their distance ranking was assigned randomly (e.g., two Tip cells at the equal minimal distance to a given Tip cell, would be randomly assigned the ranking of the closest and second closest Tip cell). We then analyzed these data for the $VEGF_{EXT}$ = 100 ng/ml experimental input which resulted in the most robust sprouting, comparing the results with the predictions of our mathematical model for the same input level that, when averaging over multiple simulations, matched the experimentally measured fraction of Tip cells. Finally, we quantified the shortest paths separating Tip cells in the equilibrated patterns (*Figure 3C* and Methods: Quantification of Tip–Tip cell distance in modeling).

The processing of experimental data described above permitted a direct comparison of modeling predictions and experimental defined Tip cells locations, First, we examined the average distances from the Tip cells to the closest, second closest, etc. neighboring Tip cells. We found that these distance distributions closely agreed with the modeling predictions, particularly for the distances up to the fourth closest neighbor, differing substantially from the predictions for the ordered 'salt-and-pepper' patterns (*Figure 3D*). A key finding was that, in agreement with the expectation from the disordered pattern model, there were frequent cases of direct contact between two Tip cells, making the average distance to the closest Tip cell around 0.5 cells in the model. In contrast, two Tip cells were always separated by at least one Stalk cell in 'traditional' 'salt-and-pepper' models. As a baseline comparison, the mathematical model with a 100-fold reduction of VEGF stimulus (1 ng/ml) exhibited a Tip–Tip distance statistics more closely comparable with the 'salt-and-pepper' models. Further analysis of the experimental distributions of Tip cell distances revealed that Tip cells were adjacent to at least one other Tip cell with 80% chance, and with at least two other Tip cells with 40% chance, and with at least three other Tip cells with 20% chance (*Figure 3E*). These Tip cell distance distributions were again in agreement with the modeling results. Taken together these findings provided strong evidence for the predicted partially disordered pattern of Tip cell specification. In our experiments, the observed cell–cell contact area varied, spanning from almost corner-to-corner contact up to approximately 50 μm. Previous studies (*Shaya et al., 2017*; *Kwak et al., 2022*) have clearly demonstrated the influence of the cell–cell contact area on NOTCH signaling, but the values get nosy in the middle range, particularly when excluding extremely low cell–cell contact areas. Reflecting these findings, we excluded the corner contacts, which might correspond to extremely low cell–cell contact areas, from the Tip–Tip distance measurements as depicted in *Figure 3B*. We also made an assumption that variations in cell–cell contact size within tens of microns correlate weakly with the strength of NOTCH signaling. This assumption did not impede our effort to compare the overall trends with results from modeling using hexagonal cells, as shown in *Figure 3D, E*.

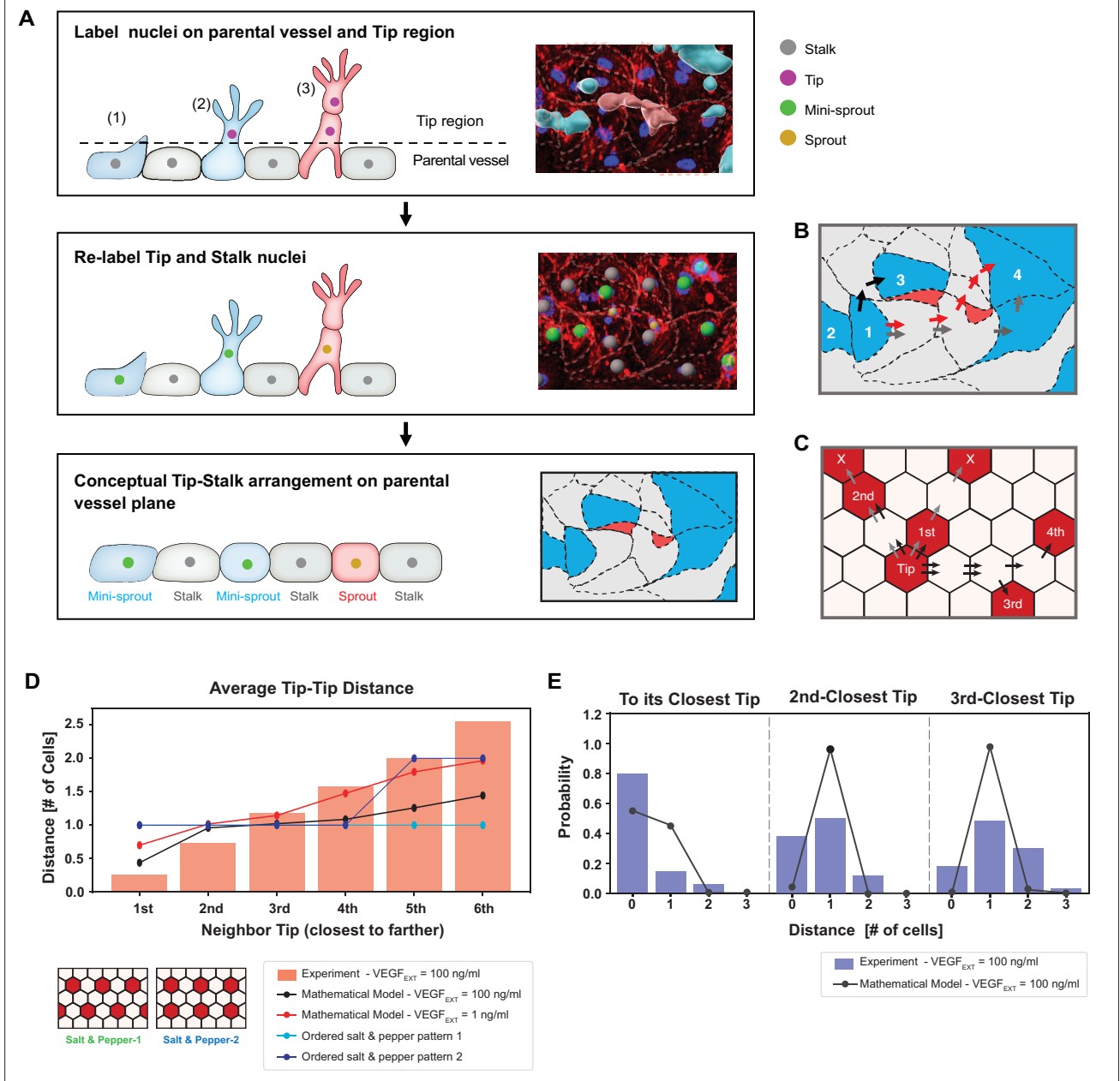

**Figure 3.** Experimentally measured spatial distribution of Tip cells defined as constituting mini-sprouts and leading sprouts is consistent with the mathematical model predictions. (**A**) Analysis pipeline to infer the 2D Tip–Stalk arrangements from 3D experimental images: experimental labeling of the nuclei of sprout/mini-sprout cells (above the plane of the parental vessel) and of the Stalk cells (below the plane of the parental vessel) is used to 'compress' the cells in each sprout or mini-sprout into a single Tip cell. Tip–Tip distance is defined as the number of cells measured in 'cell hops', that is, the minimal number of intermediate cells between randomly chosen pairs of Tip cells from experiments (**B**) (the example is identical to the inset at the bottom of (**A**)) and 2D Tip–Stalk patterns from mathematical modeling (**C**). Black arrows indicate minimal and valid cell hops between Tips, whereas gray arrows indicate minimal but invalid cell hops (passing through other Tip cells); red arrows indicate the non-minimal cell hops which does not count in the Tip–Tip distance quantification. (**D**) Tip cell distance distribution from any given Tip cell. Red bars depict experimental measurement for VEGF$_{EXT}$ = 100 ng/ml and black line depicts the model's prediction for VEGF$_{EXT}$ = 100 ng/ml. For reference, dashed lines indicate the expected Tip–Tip distance distribution of 'perfect' salt-and-pepper patterns shown in the inserts. (**E**) Detailed distance distribution for the closest Tip (left), second closest Tip (middle), and third closest Tip (right).

The online version of this article includes the following source data for figure 3:

**Source data 1.** Raw data for Tip–Tip distance measurements in **Figure 3D, E**.

## Dynamic tracking of angiogenic cell fate specification

The integrative, computational, and experimental analysis presented above suggested that the spatial Tip cell distribution can be well explained by the model of a partially disordered 'salt-and-pepper' mechanism. However, it is not clear whether all such Tip cells would spearhead the formation of a new sprout, or retract back to an alternative (Stalk) cell fate. We addressed this question by dynamically tracking the fates of mini-sprouts to examine whether this state is an intermediate step toward the sprout formation. Specifically, we imaged the progress of sprouting in the same areas of a live parental vessel at different time points of 1, 3, 7, and 28 hr of incubation in 100 ng/ml of VEGF (*Figure 4A–H*). We found that all sprouts formed either directly from Stalks or from mini-sprouts, suggesting a non-observed transition from Stalk to mini-sprout due to observational timeframe limitations. Strikingly, however, not all mini-sprouts persisted and initiated sprout formation. Instead, many mini-sprouts retracted and new mini-sprouts formed during the time-course of the analysis. We then tracked a group of 118 cells that adopted the mini-sprout phenotype at least once over a period of 28 hr after VEGF exposure. Their state change dynamics was visualized using the Sankey diagram (*Figure 4I*). The initial state of any sprouts or mini-sprouts was classified as the Stalk cell to reflect the hypothesized 'salt-and-pepper' patterning structure, entirely consisting of either Tip or Stalk cells. When a mini-sprout retracted, it was newly marked as a Stalk cell. By the final time point of 28 hr of VEGF exposure, 45.8% of the cells that displayed the mini-sprout phenotype at least once during the experiment retracted back to the Stalk state, illustrating the highly dynamic phenotype of mini-sprout extensions and retractions. Of the remaining cells, 41.5% and 12.7% were classified to be either in the mini-sprout or sprout-leading Tip cell states, respectively. Although sprout formation continued throughout the experiment, the rate of conversion of mini-sprouts to full-fledged sprouts gradually decreased over time, with 13.6%, 2.9%, and 7.5% of mini-sprouts becoming sprout-leading Tip cells in the time ranges of 1–3, 3–7, and 7–28 hr, respectively (*Figure 4K–M*). In most cases (86.7%), sprouts emerged from *newly formed* mini-sprouts (i.e., the cells that were Stalk cells and then mini-sprouts in the preceding two time points), suggesting that mini-sprouts represent transient states rapidly converting to either the fully committed sprout state or to the Stalk state (*Figure 4O*). These observations raised the question of what might define the commitment of a mini-sprout to the sprout differentiation. We next addressed this question by analyzing the spatial distribution of fully formed sprouts over the observed area of the parental vessel.

## Random uniform model accounts for spatial distribution of extending sprouts

While the NOTCH/VEGF mathematical model could quantitatively resolve Tip–Stalk spatial patterns, it did not capture the rate of cell fate switching or explained the commitment of Tip cells to lead the formation of a mature sprout. To identify the underlying principles of sprout initiation, we thus integrated the multicell model with several alternative phenomenological hypotheses (summarized in *Figure 5A–D*). We then tested these hypotheses against the measured distributions of distances between sprouts and, in particular, the observation that sprouts were always separated by at least one non-sprout cell (blue bars in *Figure 5E*). In these tests, we ensured that the results were normalized to the overall density of sprouts observed experimentally (see Method: Phenomenological models of Sprout selection). The first straightforward hypothesis was that mini-sprouts commit to the sprout phenotype independently of the location of other forming sprouts, constituting the 'cell-autonomous sprout selection' model (*Figure 5A, B*). In this case, however, the corresponding model predicted multiple contacts between sprouts (black line in *Figure 5E*), in sharp contrast with the experimental observation. The observation that most sprouts are in contact with at least another Tip (*Figure 5— figure supplement 1A*), but never in contact with another sprout, suggested a control mechanism where sprout selection inhibits nearby Tip cells from committing to the same fate. This led to two additional alternative hypotheses. In 'repulsion between sprouts' model, it was assumed that sprouts cannot be in contact; therefore, Tip cells cannot commit to the sprout phenotype if already in contact with a sprout (*Figure 5C*). In the 'random uniform' model, it was assumed that sprouts are selected randomly, but maximizing their overall spread in the lattice (*Figure 5D*; see Method: Phenomenological models of Sprout selection). While both models correctly predicted sprouts to never be in contact, the 'random uniform' model better described the cases where adjacent sprouts are separated by two or more cells (*Figure 5E*).

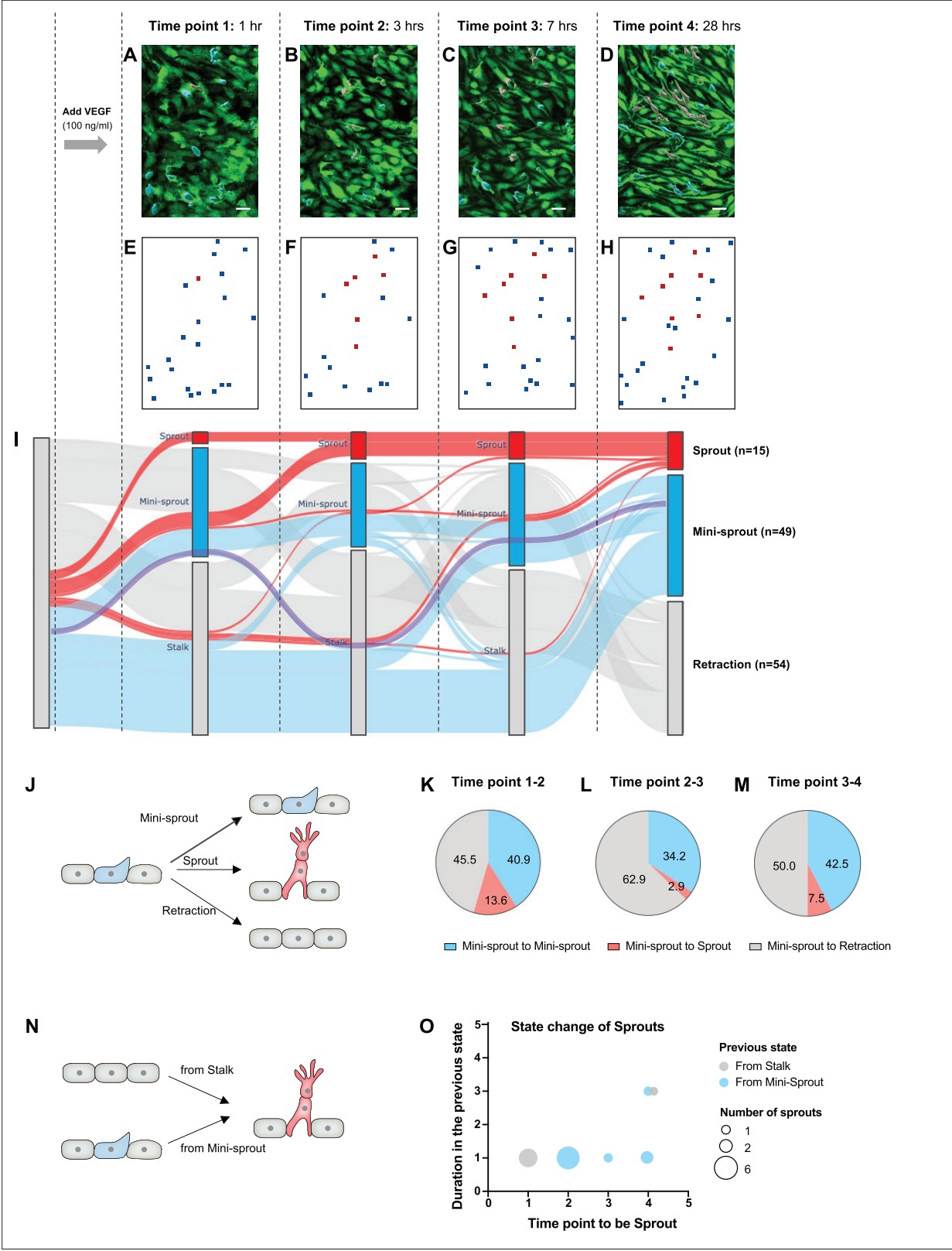

**Figure 4.** The dynamics of mini-sprout and sprout formation suggest frequent mini-sprout retractions, since only a subset of mini-sprouts becoming fully formed sprouts. Green fluorescent protein (GFP)-expressing endothelium in the 3D vessel setup captured 1 hr (**A**), 3 hr (**B**), 7 hr (**C**), and 28 hr (**D**) after 100 ng/ml of vascular endothelial growth factor (VEGF) treatment. Sprouts and mini-sprouts are identified by red and blue surface entities, respectively. Square marks representing the positions of sprouts (red) and mini-sprouts (blue) in the original images at each time point (**E–H**). (**I**) Sankey diagram

*Figure 4 continued*

demonstrating the dynamic state change of sprouts with red lines and mini-sprouts with blue lines throughout the time points. And gray lines represent mini-sprouts which ended up being retracted at the last observation, time point 4. A purple line shows an example of the state change from a Stalk (initially non-invading endothelial cell) to mini-sprout, retraction, mini-sprout, and mini-sprout at each time point. Only cells that that became mini-sprout at least once during the experiment are shown. (J) Different types of observed transitions between consecutive time points when starting from the mini-sprout state: maintain the mini-sprout state, become a sprout, or retract to the Stalk state. The ratio of states switched from mini-sprouts in the previous time point 1 (K), time point 2 (L), and time point 3 (M). (N) The two observed pathways to sprout formation between consecutive time points: direct Stalk to sprout or mini-sprout to sprout transition. Once a newly formed vessel becomes a sprout, it is permanently committed. (O) Duration of staying as a mini-sprout or a Stalk in the previous state before being committed to a sprout.

The online version of this article includes the following source data for figure 4:

**Source data 1.** Raw data tracking the dynamics of mini-sprout and sprout formation in each timeframe for *Figure 4I–O*.

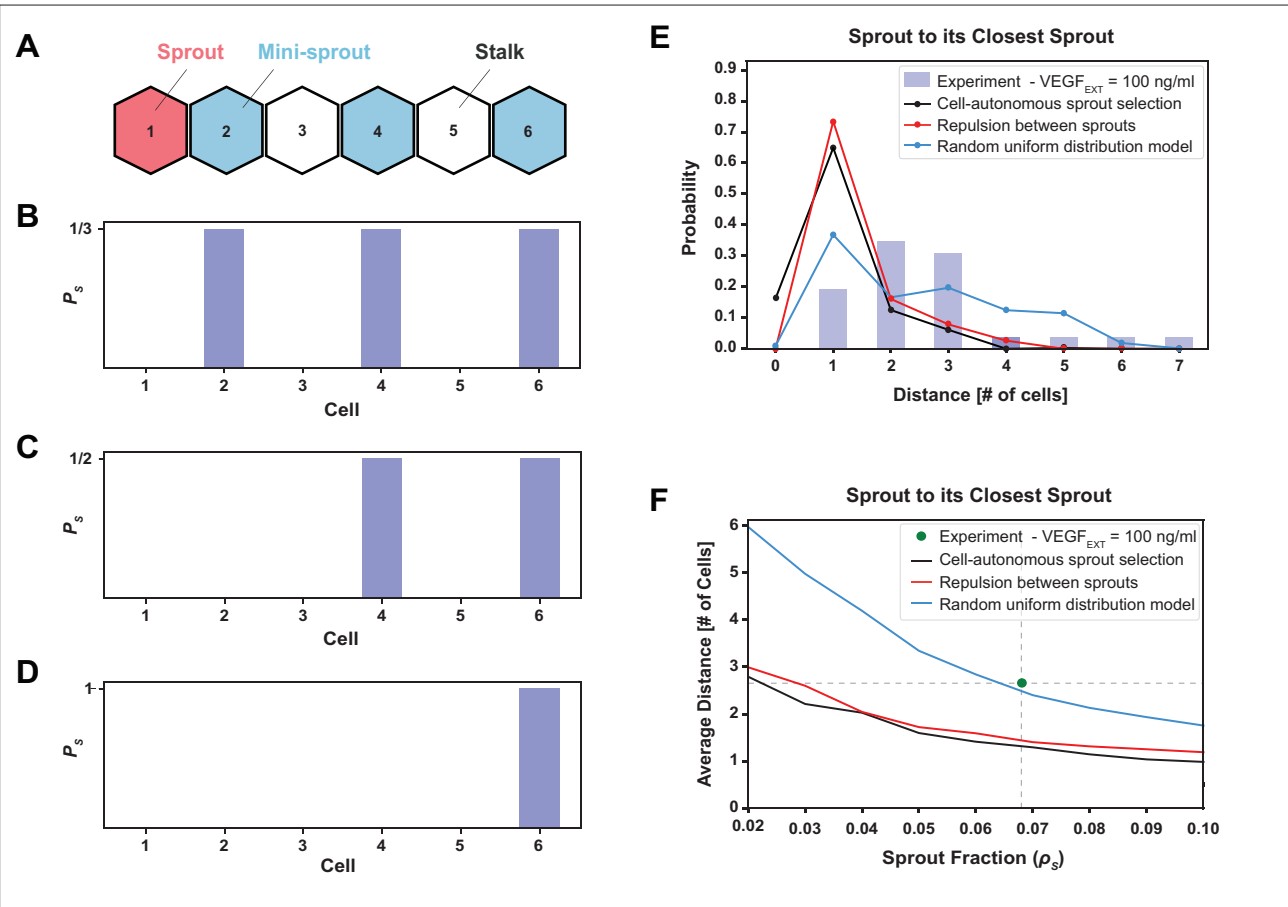

**Figure 5.** The phenomenological model favoring maximal sprout–sprout distances for a given number of sprouts (random uniform distribution) is most consistent with the experimental observations. (A) An example of one-dimensional Tip cell distribution, including a sprout and mini-sprouts, and Stalks pattern. (B) Sprout selection probability ($P_S$) for the cell-autonomous model if a new sprout was added to the pattern of (A). Stalk cells cannot become sprouts, and existing mini-sprouts share the same selection probability. (C) Sprout selection probability ($P_S$) for the sprout repulsion model. The leftmost mini-sprout cannot be selected because it is already in contact with an existing sprout, while the remaining two mini-sprouts share the same selection probability. (D) Sprout selection probability ($P_S$) for the random uniform distribution model. The rightmost mini-sprout maximizes the distance to the existing sprout and is therefore the only viable selection. (E) Sprout distance distribution to its closest sprout neighbor. Blue bars indicate experimental results for $VEGF_{EXT}$ = 100 ng/ml while black, red, and blue lines depict the three different models of sprout selection (cell-autonomous, repulsion between sprouts, and random uniform distribution, respectively). (F) Average distance between a sprout and its closest sprout neighbor in the model as a function of sprout cell fraction in the lattice for the three proposed models of sprout selection. The green dot highlights the experimental sprout fraction and distance at $VEGF_{EXT}$ = 100 ng/ml.

The online version of this article includes the following figure supplement(s) for figure 5:

**Figure supplement 1.** Phenomenological models of Sprout selection.

To test these mechanisms more rigorously, we computed the average distance (in cell numbers) between pairs of closest sprouts while also varying the number of sprouts allowed in the lattice (*Figure 5—figure supplement 1B–D*), thus generating a curve of the typical sprout–sprout distancing as a function of sprout density in the lattice (*Figure 5F*). The 'random uniform' model predictions agreed very closely with the experimentally observed combination of sprout fraction and sprout–sprout distance, whereas the other two models greatly underestimated the distances between sprouts (*Figure 5F*). Furthermore, while all models overestimated the fraction of the adjacent sprouts that are one cell away from the current sprout and underestimated the fractions of sprouts at greater distances, the deviation of the 'random uniform' model predictions for this inter-sprout distribution was the lowest of all the models, again supporting the 'random uniform' model as the more likely to account for sprout selection (*Figure 5F*).

## Fibronectin distribution may mediate sprout induction

Random distribution maximizing the distance between sprouts is similar to allelopathy models, accounting to spatial dispersion of species maximizing distance between them. In these models, the key postulated mechanism is inhibition of growth of individuals of the same species through a mutual suppression mechanism (*Liu, 2003*; *Fraenkel, 1959*; *Willis, 2007*). Another analogous set of mechanisms are embedded within the concept of the Turing pattern formation, the key to which is diffusible negative feedback regulator setting spatial distribution of morphogenic features (*Turing, 1990*; *Maini et al., 2006*). A variant of such mechanisms is a model postulating depletion of some ingredient that is key to the local growth, by its active redistribution toward the growing pattern features and depletion from the zones between them, rather than active mutual inhibition of the pattern forming units.

Given these prior models, we hypothesized that a similar mechanism may account for the dispersion patterns of sprouts. We focused on the ECM as a possible medium accounting for the positive and negative pattern-setting interactions. Indeed recently, it has been observed that collagen can be re-organized by the growing sprouts, so that it is concentrated around the extending sprouts and depleted elsewhere (*Feng et al., 2013*; *Senk and Djonov, 2021*; *Kirkpatrick et al., 2007*). We explored whether a similar distribution is also be observed for fibronectin, an ECM component that is critical for the formation of lumenized sprouts (*Alon et al., 1994*; *Wijelath et al., 2006*; *Astrof and Hynes, 2009*; *Bayless et al., 2000*). Fibronectin expression levels in the vicinity of individual cells comprising the parental vessel and the emerging sprouts was experimentally assessed by immunostaining, with simultaneous cell identification using induced cytoplasmic GFP expression and nuclear staining, followed by 3D reconstruction (*Figure 6A–H*). Untreated quiescent cells (No treatment) and mini-sprouts showed similar levels of fibronectin expression (*Figure 6H*). Interestingly, the fibronectin expression was highly enriched at the base but not the Tip areas of the extending sprouts, suggesting that it may be a key determinant of sprout induction but not extension (*Figure 6B, C, H*). We also examine the regional variation of fibronectin expression in larger areas, which was less variable and potentially more relevant to sprout extension. In particular, we accessed the ratio of cells having fibronectin levels higher than a threshold in groups of seven cells around sprouts and mini-sprouts (*Figure 6I, J*). The overall expression levels when all region types were combined was relatively uniform (*Figure 6J*), third panel. Strikingly however, regions around sprouts (*Figure 6J*, first panel) showed oppositely skewed patterns vs. mini-sprouts (*Figure 6J*, second panel). Specifically, the fibronectin expression levels around sprouts were higher than the threshold, whereas the fibronectin levels around mini-sprouts were lower than the threshold. Altogether, these results supported the model in which fibronectin can indeed serve as a mediator of Turing-like induction of sprouting patterns, through re-modeling that enriches it at the points of sprout induction and depletes it at the points where Tip cells (mini-sprouts) are not stabilized to form full-fledged, lumenized sprouting bodies.

## Discussion

A major challenge of the analysis of tissue development and homeostasis is understanding of how differentiation into distinct cell types can be robustly achieved, while also being sensitive to various pro-differentiation and morphogenic cues and, potentially affected by molecular noise in biochemical reactions. In the context of angiogenesis, this challenge more specifically relates to enabling effective vascular morphogenesis through robust yet environmentally responsive differentiation of endothelial

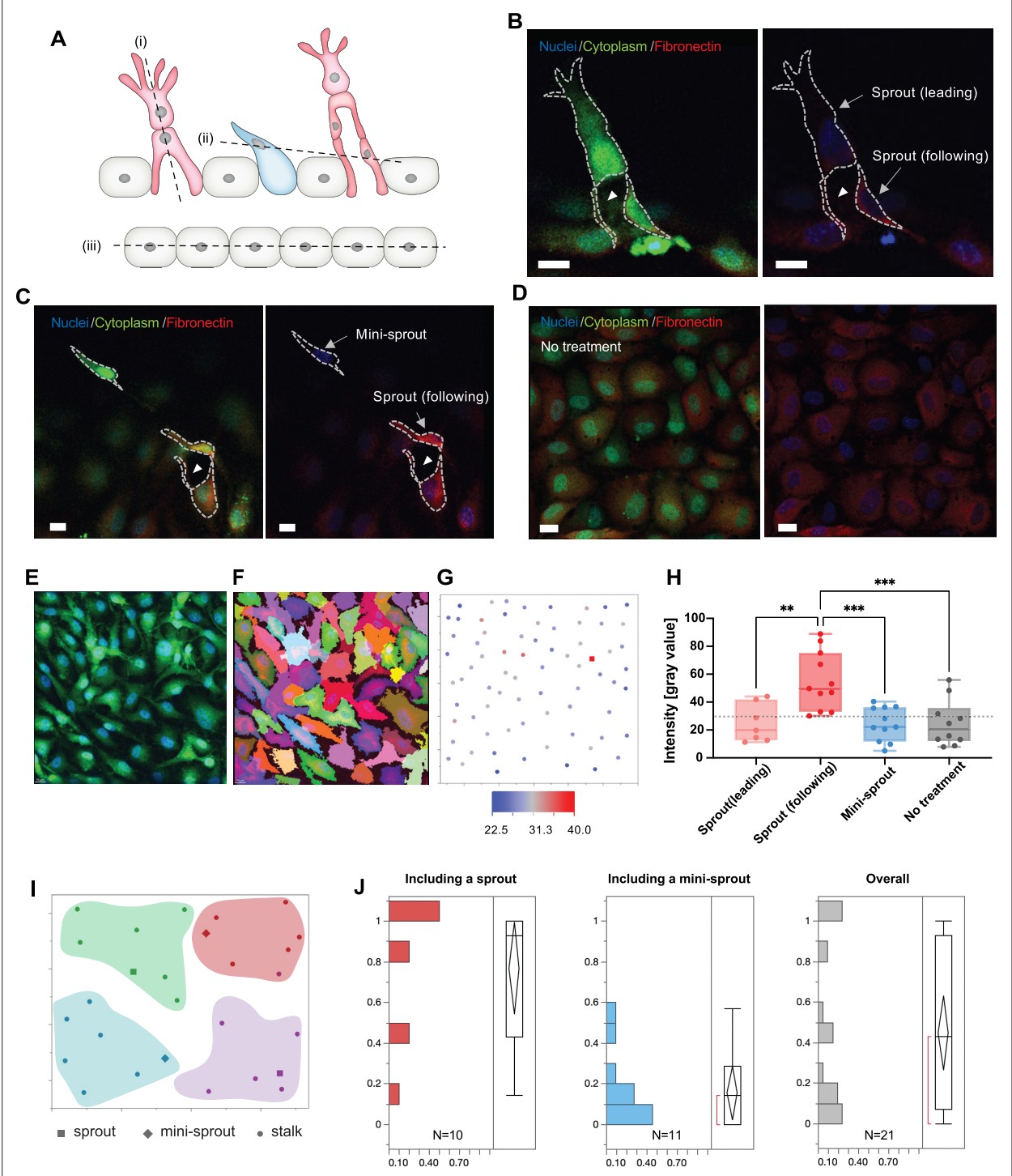

**Figure 6.** Fibronectin distribution on parental and newly formed vessels reveals preferred distribution at the bases of sprouts. (**A**) A schematic describing cross-sectional planes for subsequent confocal images: (i) for (**B**), (ii) for (**C**), and (iii) for (**D**). (**B**) Localization of fibronectin expression to a following cell than in a leading cell in a sprout. (**C**) Higher fibronectin expression in a sprout (a 'following' cell at the base of the sprout) than at a mini-sprout. (**D**) Intrinsic heterogeneity of fibronectin expression on quiescent endothelium displaying no mini-sprout or sprout formation. Images are 3D reconstructions of confocal z-stacks. Scale bars: 15 μm. Cells on the parental vessel were identified by GFP expression in the cytoplasm (**E**), then segmented (**F**). (**G**) Fibronectin intensity of each cell on the parental vessel is marked as a dot at the corresponding *x* and *y* positions of the cell centroids. Fibronectin intensities for sprouts (following cells) or mini-sprouts are indicated as squares. (**H**) Fibronectin intensity of leading cells and following cells of sprouts, mini-sprouts, and quiescent cells. Data are presented by box and whiskers with all individual points representing each cell ($N_{sprout(leading)}$: 7; $N_{sprout(following)}$: 11; $N_{Mini-sprout}$: 11; $N_{No\ treatment}$: 10). One-way ANOVA analysis was performed followed by Tukey's multiple comparisons test

*Figure 6 continued on next page*

*Figure 6 continued*

(\*p < 0.05, \*\*p < 0.01, \*\*\*p < 0.001, \*\*\*\*p < 0.0001). (**I**) Cellular layer was segmented into groups containing seven neighboring cells to assess the local environment for each group. (**J**) Distributions of the ratio of cells having fibronectin levels higher than a threshold, the minimum value of sprout in (**H**), in a group of seven neighboring cells defined in (**I**), which included either a sprout or a mini-sprout. The overall distribution covers both regions.

The online version of this article includes the following source data for figure 6:

**Source data 1.** Raw fibronectin intensity data for *Figure 6H, J*.

cells into Tip and Stalk cells states. This process is guided by the multiple pro-angiogenic cues, such as VEGF and by the local tissue organization and mediated by the paracrine Delta–NOTCH cell signaling. Recent mathematical models of this process (*Galbraith et al., 2022*; *Koon et al., 2018*) have considered the robustness of the Delta–NOTCH differentiation to molecular noise, concluding that noisy biochemical reactions can both disrupt and enhance the spatial differentiation patterns, depending on the magnitude and spatial distribution (*Galbraith et al., 2022*). The effect of pro-angiogenic cues, such as VEGF on robustness of spatial differentiation patterning remains substantially less explored. Furthermore, compiling between modeling predictions and experimental validation has been challenged by the complexity of in vivo angiogenesis analysis, both on cellular and molecular levels.

Here, we integrated engineered experimental angiogenic assay and a spatially resolved computational modeling analysis to explore the spatially and temporally resolved effects of VEGF on angiogenic cell specification. Our results suggest that VEGF can have a dual role in inducing the initial Tip–Stalk cell differentiation. On the one hand, a low level of exogenous VEGF is essential to induce Delta–NOTCH signaling and the classical ordered 'salt-and-pepper' pattern, with approximately 25% of the cells adopting the Tip cell fate, as expected (*Collier et al., 1996*). This role of VEGF is conceptually similar to the classical case of EGF induced NOTCH signaling in *C. elegans* vulva development (*Shin and Reiner, 2018*; *Yoo et al., 2004*). On the other hand, our results also suggest that an increase of VEGF levels can introduce disorder into this pattern, similar to order–disorder transition for various composite materials (*Yang et al., 2017*; *Bu et al., 2022*), which may occur with increasing temperature and have the properties of a sharp phase transition. More specifically, high VEGF levels may play the role similar to an increased temperature in order–disorder transitions, leading to emergence of partially disordered 'salt-and-pepper' structures. These disordered structures are characterized by higher than expected fractions of Tip cells and, consequently, an increased occurrences of otherwise disallowed adjacent Tip cells. Importantly, for all VEGF input levels and the resulting spatial patterns, the Tip–Stalk cell specification continued to be robust, although the degree of induction of NOTCH signaling was dependent on the VEGF dose. These results suggested that cell specification patterns deviating from the expected 'salt-and-pepper' one can develop not only due to noisy NOTCH signaling, as suggested by prior models, but also due to the control of the fractions of alternative cell states by the magnitude the pro-differentiation cue.

In spite of the observation that disordered 'salt-and-pepper' patterns still display robust differentiation, the spatial patterns of cell co-localization can generate inherent instabilities, for example, due to two adjacent Tips mutually suppressing their fate selection through NOTCH signaling. Instabilities of this sort may increase the sensitivity of the differentiation process to additional cues and can also lead to facilitated dynamic switching of cell fates (e.g., Tip to Stalk and vice versa) over prolonged periods of time. Such instabilities might lead to oscillatory-like fluctuations of NOTCH signaling as observed in other differentiation processes (*Zhang et al., 2021*; *Venzin and Oates, 2020*; *Kageyama et al., 2010*) and in endothelial cell sheets under high VEGF concentration inputs (*Ubezio et al., 2016*). Importantly, these instabilities may also underlie the striking observation of the continuous retraction and extension of mini-sprouts (protruding Tip cells) observed in our study. This dynamic fate-switching behavior can thus represent a signature of an unstable differentiation process that may either stabilize, in response to additional cues, leading to a specific morphogenetic outcome, such as the extension of a stable sprout, or display a prolonged instability resulting in a lack of pronounced morphogenesis. Such poised but unstable states may be similar to the undifferentiated state of neurogenic progenitors

displaying oscillatory NOTCH signaling, which can proceed to differentiation after the oscillation is resolved into a temporally stable NOTCH activity (*Shimojo et al., 2008*).

The cues stabilizing cellular differentiation and morphogenesis can vary and represent the signature of both global and local pro- and anti-angiogenic environments. For instance, our prior analysis indicated that exposure of model parental vessels to VEGF only rather than a more complete pro-angiogenic cocktail of various factors, can result in formation of mini-sprouts (and hence, effective Tip–Stalk cell differentiation), but not full-fledged sprouts (*Kang et al., 2019*). Sprout formation may also be modulated by the presence of mural cells and pro-inflammatory cytokines, which can indirectly modulate the NOTCH activity, but could also have additional effects, serving to stabilize a specific differentiation outcome. However, even if sprouts do form, it is not clear how their spatial distribution may arise and be potentially controlled by the environmental inputs. Our results argue that, for a given number of sprouts forming in the parental vessel, their mutual distances are maximized. This spatial distribution is consistent with a Turing-like mechanism (*Turing, 1990*; *Maini et al., 2006*), implying the existence of a long-distance interaction inhibiting formation of new sprouts in the vicinity of the existing ones. Although the actual mechanism of putative Turning-like pattern formation is not fully elucidated here, our results are consistent with a variant of this regulatory behavior, in which a component of the ECM, fibronectin is actively redistributed by the nascent sprouts, with this ECM component being enriched at the points of sprout formation, but depleted in other zones, thus preventing sprout induction in the depleted areas. This redistribution is indeed equivalent to the classical Turing mechanisms that would involve generation of an explicitly inhibitory compound by the growing sprouts. Fibronectin has been implicated as a key pro-angiogenic ECM component, possibly due to its integration with VEGF, which further increases the plausibility of this mechanism (*Wijelath et al., 2006*). Interestingly, we find elevated levels of fibronectin at the bases of extending sprouts, but not at their Tips. This finding has two implications. First, fibronectin may be an important factor in the sprout induction rather than extension, hence the growing sprout can progress into the surrounding matrix beyond the area of enriched fibronectin, leaving it behind. Secondly, fibronectin can also promote lumen formation (*Astrof and Hynes, 2009*; *Bayless et al., 2000*), thus its enrichment at the bases of extending sprouts can further contribute to their lumenized structure. Furthermore, this mechanism can help explain the dramatic influence of cytokines, such as TNF, in preventing sprout formation, which can happen in sharp, TNF dose-dependent manner (*Kang et al., 2019*). Indeed, TNF avidly binds to fibronectin (*Alon et al., 1994*) and therefore, at high enough concentrations, would have a particular anti-angiogenic effect in the areas of increased fibronectin density, which according to the proposed mechanism would be the areas of incipient sprouts. This would dramatically increase its anti-sprouting effect, even if the effect of this cytokine on the NOTCH-dependent cell specification is more muted (*Kang et al., 2019*). Of note, another version of the Turing mechanism has recently been suggested to account for the branching of the sprouts, also involving formation of new Tip cells leading individual branches, although the molecular mechanism postulated in that analysis was distinct from the one proposed here (*Guo et al., 2021*; *Xu et al., 2017*). These models, though plausible, will need to be further tested to ascertain causality of the proposed mechanisms, although at this stage the Turing-like mechanisms appear to be the best candidates to explain the experimental results we obtained.

Overall, our analysis supports the following dynamic view of angiogenic induction. The VEGF input can induce NOTCH signaling and formation of Tip cells that can behave as mini-sprouts. At lower VEGF input, a more ordered pattern of Tip cell induction can lead to formation of approximately 25% of Tip cells, but relatively few of these will become sprouts, reflecting lower sensitivity to additional ambient cues, promoting sprout formation. On the other hand, at higher VEGF inputs, a greater disorder of Tip cell patterning permits higher sensitivity to external cues, such as fibronectin, that can stabilize the sprouting. In addition, a more frequent co-localization of Tip cells under this condition enables initiation of a sprout from two adjacent cells, as observed in a fraction of cases in our experiments (the case in *Figure 1—figure supplement 1* involving formation of a sprout by two adjacent cells). The initial emergence of sprouts leads to a progressively less likely sprout formation and to the overall maximization of the distance between the sprouts. Both these observations are consistent

with non-local inhibition of sprout formation around the sprouts that have already formed. A plausible mechanism for this inhibition and the overall pattern formation is the redistribution of fibronectin from the zones between sprouts toward the incipient sprouts, constituting both a positive and negative feedback loops, commonly assumed in a variant of the Turing patterns from action. This mechanism is very sensitive to various local inputs distinct from VEGF and fibronectin, which can further influence the location and density of the developing blood vessels. These may include the effects of pericytes or local inflammatory environments, as mentioned above, but also other ECM components, such as collagens, that may enhance the protrusion of Tip cells and thus stabilize the emergent sprouts both directly and indirectly. Endothelial fate induction process may be interesting to contrast with other complex multicellular processes, including collective epithelial migration, where NOTCH signaling similarly modulates fine-grained patterns of leader and follower cells (*Vilchez Mercedes et al., 2021*), and underscore the need in the future to develop more refined models that explicitly integrate the interconnections between biochemical and mechanical regulation of Tip–Stalk fate (*Stassen et al., 2020*). The results in this study can further inform our understanding of angiogenesis in physiological and patho-physiological conditions. In particular, in many circumstances, the levels of VEGF are determined by the degree of hypoxia, which can be highly elevated following oxygen supply interruption, for example, in wound healing or ischemia, or due to progression of neoplastic growth. Our results suggest that in these cases, formation of sprouts can be dysregulated due to higher incidences of co-localizations of prospective Tip cells. In addition, since these conditions are frequently accompanied by altered synthesis of ECM, the sprout density can increase, which may lead to formation of denser and less developed vascular beds frequently observed as a result of tumor angiogenesis (*Ruoslahti, 2002*; *Chung et al., 2010*). Our results thus suggest that the disorder and higher plasticity of the endothelial cell fate speciation at higher VEGF inputs can be a key contributor to some pathological states associated with persistently hypoxic conditions.

The analysis presented here highlights the utility of combining experimental engineered vasculature models with mathematical analysis as integrated research platforms to gain a progressively better insight into angiogenesis in highly controlled micro-environments. Although, questions remain about the mechanistic underpinnings of the phenomena observed here and in related studies, the analysis enabled by these model systems can help formulate guiding hypotheses for further understanding of angiogenesis in vivo, while decoupling the complexities of the native angiogenic environments. We anticipate that the dynamic exploration presented here can help pave the way for further quantitative understanding of this key biological process.

# Materials and methods

**Key resources table**

| Reagent type (species) or resource | Designation | Source or reference | Identifiers | Additional information |
|---|---|---|---|---|
| Antibody | Recombinant Anti-Fibronectin antibody (Rabbit monoclonal) | abcam | ab268020 | IF (1:50) |
| Antibody | Goat anti-Rabbit IgG (H+L) Highly Cross-Adsorbed Secondary Antibody, Alexa Fluor Plus 594 (Goat polyclonal) | Invitrogen | A32740 | IF (1:100) |
| Peptide, recombinant protein | Animal-Free Recombinant Human FGF-basic (bFGF) | PeproTech | AF10018B | |
| Peptide, recombinant protein | VEGF Recombinant Human Protein | Life Technologies | PHC9394 | |
| Chemical compound, drug | Sphingosine-1-phosphate | Sigma-Aldrich | S9666 | |
| Chemical compound, drug | Phorbol myristate acetate | Sigma-Aldrich | P1585 | |
| Biological sample (*Homo sapiens*) | Primary human brain microvascular endothelial cells transfected with GFP-expressing lentivirial particles. | Angio-Proteomie | cAP-0002GFP | Isolated from normal human brain tissue |
| Software, algorithm | IMARIS 9.8.0 | Bitplane | | https://imaris.oxinst.com |

*Continued on next page*

*Continued*

| Reagent type (species) or resource | Designation | Source or reference | Identifiers | Additional information |
|---|---|---|---|---|
| Software, algorithm | PRISM 9.0.0 | GraphPad | | https://www.graphpad.com/features |
| Software, algorithm | Jupyter Notebook 6.1.6 | Project Jupyter | | https://jupyter.org/install |
| Software, algorithm | Python 3.12.1 | Python Software Foundation | | https://www.python.org |
| Other | Hoechst 33342 | Thermo Fisher | H3570 | IF (5 µg/ml) |

## Fabrication of 3D vessel chip

Chips for mimicking 3D angiogenesis in vitro were fabricated as introduced in our previous work (*Kang et al., 2019*). The chip for live-cell imaging consists of a polydimethylsiloxane (PDMS) chamber, an engineered blood vessel embedded in collagen gel, and the whole construct was placed on a glass-bottom dish (*Figure 1A*). A single line mold (D: 200–250 µm, L: 10 mm) of polylactic acid (PLA) which has a semicircular cross-section was deposited on a Petri dish with a 3D printer (Ultimaker). After a PDMS chamber was put on the mold, pre-mixed collagen solution (5 mg/ml, Type 1 collagen, BD) according to the manufacturer's protocol was injected through a hole on the PDMS chamber and collagen polymerization was induced on ice for 30 min and at 37°C for 1.5 hr. The PDMS chamber including the collagen construct was carefully peeled off, then the bottom side was sealed with a glass bottom of a dish. Laminin solution (60 µg/ml in phosphate-buffered saline, Sigma-Aldrich) was injected into the channel pre-made in collagen, then the chip was flipped upside down and incubated at 37°C for 1 hr. Finally, endothelial cell suspension ($1 \times 10^7$ cells/ml) was injected into the channel, then the chip was flipped upside down again and incubated at 37°C for 1 hr to allow cells to attach to the lumen. Once endothelial cells form a confluent monolayer on the channel surface, a fresh medium was injected into the engineered blood vessel and changed every 12 hr before the treatment with pro-angiogenic factors.

## Cell culture

GFP-expressing human brain endothelial cells (GFP-HBMEC) were purchased from Angio-Proteomie. According to the information provided by Angio-Proteomie, primary human brain endothelial cells, isolated from normal human brain tissue, were transfected with GFP-expressing lentiviral particles and subsequently selected for puromycin resistance. Cells were cultured in a growth medium of M199 (Gibco) supplemented with 20% fetal bovine serum (Life Technologies), 1% HEPES buffer (Thermo Scientific), 1% Glutamax (Thermo Fisher), 1% antibiotic–antimycotic (Thermo Fisher), Heparin (25 mg/500 ml, Sigma-Aldrich), and endothelial cell growth supplement (Sigma-Aldrich) and used for further experiments at passage 9.

## Induction of angiogenesis

For inducing angiogenesis in the 3D vessel chip, the growth medium was supplemented with 40 ng/ml of basic fibroblast growth factor (Thermo PeproTech), 500 nM of Sphingosine-1-phosphate (S1P, Sigma-Aldrich), and 75 ng/ml of phorbol myristate acetate (Sigma-Aldrich). And 100 or 10 ng/ml of VEGF was added on the top reservoir of the 3D vessel chip to create a VEGF gradient toward the engineered blood vessel (*Figure 1B*).

## Mathematical model of VEGF/NOTCH signaling

We generalize existing mathematical models of the interconnected signaling between the NOTCH and VEGF pathways to a two-dimensional multicellular scenario (*Kang et al., 2019*; *Boareto et al., 2015a*). The temporal dynamics of NOTCH (N), Delta (D), Jagged (J), NOTCH intracellular domain or NICD (I), and VEGF receptor ($V_R$) in a cell are modeled with ordinary differential equations:

$$\frac{dN}{dt} = N_0 H^{+(I)} - k_T N(D_{EXT} + J_{EXT}) - k_C N(D+J) - \gamma N \tag{1a}$$

$$\frac{dD}{dt} = D_0 H^{-(I)} H^+(V_R V_{EXT}) - k_T N_{EXT} D - k_C ND - \gamma D \tag{1b}$$

$$\frac{dJ}{dt} = J_0 H^+(I) - k_T N_{EXT} J - k_C NJ - \gamma J \tag{1c}$$

$$\frac{dI}{dt} = k_T N(D_{EXT} + J_{EXT}) - \gamma_I I \tag{1d}$$

$$\frac{dV_R}{dt} = V_{R0} H^-(I) - k_T V_R V_{EXT} - \gamma V_R \tag{1e}$$

NOTCH, Delta, Jagged, and VEGF receptor are produced with basal rates $N_0$, $D_0$, $J_0$, $V_{R0}$, and are degraded with rate constant $\gamma$. The basal production rates are modulated by NICD that transcriptionally activates NOTCH and Jagged while inhibiting Delta and VEGFR via shifted Hill functions:

$$H^S(I, I_0, n, \lambda) = \frac{1 + \lambda \left(\frac{I}{I_0}\right)^n}{1 + \left(\frac{I}{I_0}\right)^n} \tag{2}$$

where $I_0$ is a threshold NICD level, $n$ is the Hill coefficient, and $\lambda$ represents the target's production rate fold-change at high NICD concentrations ($I \gg I_0$). Therefore, $\lambda > 1$ implies transcriptional activation while $\lambda < 1$ implies transcriptional inhibition. For brevity, activating and inhibiting Hill functions are denoted by $H^{+(I)}$ and $H^{-(I)}$, respectively.

Receptors and ligands can bind to external ligands/receptors with binding rate constant $k_T$. In the case of NOTCH signaling, $N_{EXT}$, $D_{EXT}$, and $J_{EXT}$ represent the average levels of NOTCH, Delta, and Jagged in the six nearest neighbor cells on the hexagonal lattice. Conversely, VEGF is modeled as an external signal provided to all endothelial cells; therefore, all cells are exposed to the same fixed level ($V_{EXT}$). Moreover, NOTCH receptors and ligands can bind within the same cell with a rate constant $k_C$, which results in the degradation of the ligand–receptor complex without any downstream signaling (cis-inhibition). NICD is released upon binding of NOTCH receptors with external ligands and degraded with rate constant $\gamma_I$. Finally, VEGF receptors can bind to external VEGF ligands, thus creating activated VEGF receptor ($V_R V_{EXT}$), which in turn inhibits the production of Delta. Details on parameter values are presented in *Table 1*. Compared to previous models, we rescaled the Hill function threshold for VEGF-mediated activation of Delta in order to match quantitatively the ratio of Tip/Stalk cells between model and experiment when the external input is $V_{EXT} = 100$ ng/ml.

In this project, we focus specifically on the regulation of VEGF on the NOTCH–Delta signaling pathway (as shown in *Figure 2A*). A strong NOTCH–Jagged interaction can suppress Tip–Stalk differentiation and instead lead to a hybrid Tip/Stalk phenotype, which is beyond the scope of the current study. For completeness, we maintained the integrity of the entire circuit structure including NOTCH–Jagged interactions (*Equation 1c*) but fixed the production rate of Jagged at a low level that does not interfere with the bistable behavior of the VEGF/NOTCH circuit.

**Table 1.** Parameter values for simulation.

| Parameter type | Parameter | Value | Units |
|---|---|---|---|
| Production | $N_0$, $D_0$, $J_0$, $V_{R0}$ | 1200, 1000, 800, 1000 | Molecule/hr |
| Degradation | $\gamma$, $\gamma_I$ | 0.1, 0.5 | 1/hr |
| Binding | $k_T$, $k_C$ | $2.5 \times 10^{-5}$, $5 \times 10^{-4}$ | 1/(molecule hr) |
| Hill threshold | $I_0$, $V_0$ | 200, 80* | Molecules |
| Fold-change | $\lambda_N$, $\lambda_{I,D}$, $\lambda_{V,I}$, $\lambda_J$, $\lambda_V$ | 2, 0, 0, 2, 2 | Dimensionless |
| Hill coefficient | $n_N$, $n_{I,D}$, $n_{V,I}$, $n_J$, $n_V$ | 2, 2, 2, 5, 2 | Dimensionless |

*Rescaled from previous model to match experimental observation.

## Definition of Tip cells in the model

Tip cells — usually defined as high Delta, low NOTCH cells — are defined in this model as cells with Delta levels larger than $10^3$ molecules. This definition is motivated by observing that the distribution of cellular Delta at steady state in the multicell model is always bimodal irrespectively of the level of external VEGF input (see *Figure 2B*), with a large separation between the population of cells with low Delta levels (the Stalk cells) and the population of cells with high Delta levels (the Tip cells). This phenomenological definition suffices here due to the deterministic nature of the model; otherwise, more complex approaches would be necessary in presence of stochastic fluctuations (*Galbraith et al., 2022*).

## Simulation details

All results are calculated on a 30 × 30 hexagonal lattice with periodic boundary conditions. Initially, the lattice is prepared with randomized initial conditions where the initial levels of $N$, $D$, $J$, $I$, and $V_R$ within each cell are sampled from uniform distributions. Afterwards, the lattice equilibrates for 100 hr without any VEGF input to simulate the seeding of endothelial cells before VEGF is provided.

## Quantification of Tip–Tip distance in experiments

Fluorescence microscopy images of Hoechst 33342/phalloidin staining from our previous work (*Kang et al., 2019*), two images for each VEGF condition, were used to quantify the distance between Tips. Before counting Tip–Tip distance, images were pre-processed to clean up intricate 3D images as depicted in *Figure 3A* with IMARIS (Bitplane). First, all newly formed vessels were separated from parental vessels, identified as surface entities, and labeled with different colors depending on lumen formation: red for sprout and blue for mini-sprout. And all nuclei were detected as spots (*Figure 1A*, label nuclei on the parental vessel and Tip region). In this step, nuclei on the parental vessel region were marked as Stalks and the nuclei in sprout or mini-sprout surfaces were marked as Tips. Second, the nuclei of Stalks on the parental vessel region were re-examined (*Figure 3A*), Re-label Tip and Stalk nuclei. If they are connected to sprout or mini-sprout surfaces (*Figure 3A*; *Adams and Alitalo, 2007*; *Potente et al., 2011*), they were re-labeled as nuclei of sprouts or mini-sprouts and all other nuclei in the sprout or the mini-sprout were deleted. If a nucleus of a sprout or a mini-sprout is not in the parental vessel, the closest nucleus to the parental vessel was labeled as a sprout or a mini-sprout and all other nuclei were deleted (*Figure 3A*). By going through these steps, we could derive 2D conceptual Tip–Stalk arrangements from 3D images, which are comparable to 2D Tip–Stalk patterns from mathematical modeling. The pre-process is based on our assumption that the Stalk cells found in the extending sprouts emerge de novo through cell proliferation, rather by Stalk cell migration from the parental vessel.

In this study, we defined the distance between two Tips as the number of cells measured in 'cell hops', that is, the minimal number of intermediate cells between randomly chosen pairs of Tip cells. For example, in *Figure 3B*, the distances from Tip 1 to Tip 2 and to Tip 3 are 0 and 1, respectively. From Tip 1 to Tip 4, the distance is 3 in the minimal cell hops marked with gray arrows which pass through another Tip. Another cell hops marked with red arrows does not include any Tips, but the distance is 4. In this case, we discarded Tip 4 in the quantification of Tip–Tip distance from Tip 1. The same rule was applied in analyzing Tip–Tip distance from mathematical modeling (*Figure 3C*).

## Quantification of Tip–Tip distance in modeling

Upon complete lattice equilibration, cells naturally separate into two distinct groups based on low or high expression of Delta irrespectively of the external VEGF input (see *Figure 2B*). Therefore, high-Delta cells are labeled as Tips and low-Delta cells are labeled as Stalk. The algorithm to compute distances between Tip cells on the hexagonal lattice follows the following two steps: (*Adams and Alitalo, 2007*) Computing the shortest path between a given pair of cells on the hexagonal grid; and (*Potente et al., 2011*) filtering out the measurement if the cell pair is 'shielded' by another Tip cell. First, the shortest path is defined as the minimum number of intermediate cells that connect the two cells of interest. Considering two cells with coordinates $(x_1, y_1)$ and $(x_2, y_2)$, their distance has different values based on the cells relative position. In the simple case where $dx = 0$ or $dy = 0$, the distance is $d = dx + dy$. Else, if the distance is $d = dx + dy - 1$ (*Adams and Alitalo, 2007*) if $dy$ is even of *Potente et al., 2011* if $dy$ is odd but $y_1$ is even. In any other case, the distance is $d = dx + dy$. Next, to maintain

consistency with the experimental statistics, the distance is not included in the Tip cell distribution if there is at least one intermediate Tip cell 'shielding' the two Tip cells of interest. In the experimental protocol, an intermediate cell 'shields' the two Tip cells if the straight line connecting the centers of the two cells passes through the shielding cell. Therefore, we search the intercept between the line connecting the pair of Tip cells and a circle centered at the position of the shielding cell with radius of 0.5. Assuming a circular geometry for the shielding cell removes artifacts emerging due to the polygonal shape of cells in the hexagonal lattice. The sprout–sprout distance statistics presented in *Figure 5E* are similarly computed by applying the same algorithm to the set of Sprout cells in the lattice (see the following section for details of sprout selection).

## Phenomenological models of sprout selection

We developed three separate models of sprout selection: 'Cell-autonomous', 'Repulsion', and 'Random uniform distribution'. For all three cases, cells equilibrate in the hexagonal lattice. Then, Tip cells are defined based on high-Delta expression. This analysis produces a discrete snapshot of Tip and Stalk cells on the hexagonal lattice. When comparing the model prediction to the 100 ng/ml VEGF dosage experiment, we constrain the sprout selection models to reproduce the experimental sprout fraction. In the 'Cell-autonomous' model, a fraction of the Tip cells is randomly selected and defined as Sprouts independently from the phenotype of their neighbors. In the 'Repulsion' model, a fraction of Tip cells is randomly selected as Sprouts with the additional constraint that Sprouts cannot be in direct contact. To implement this constraint, we iteratively select Tips one at a time. If the selected Tip is not already in contact with a previously selected Sprout, it is promoted to the Sprout state; otherwise, a new Tip cell is selected. The iteration stops when the target number of Sprouts is reached. Finally, in the 'Random uniform distribution' model, Tips are selected as Sprouts to maximize their overall spread in the lattice. To implement this constraint, we first select a random Tip and promote it to the Sprout state. Then, the furthest Tip from the newly selected Sprout is selected and promoted to the Sprout state as well. Afterwards, a max-distance function is defined as the sum of pairwise distances between all Sprouts already selected in the lattice. At each following iteration, the Tip cell that maximizes the distance function is selected as a Sprout. The iteration stops when the target number of Sprouts is reached.

## Sprout/mini-sprout tracking and analysis

3D vessel chips were kept in a $CO_2$ incubator at 37°C and taken out only for imaging at the time point depicted in *Figure 4A–D*. GFP-expressing endothelial cells on a chip at the same position were imaged with a ×20 water-immersion objective attached to Lecia scanning disk confocal microscope. IMARIS (Bitplane) was used to separate all newly formed vessels from the parental vessel and to label them with different colors depending on lumen formation: sprout and mini-sprout. The positions of sprouts and mini-sprouts were simply marked with red and blue squares, respectively (*Figure 4E–H*). Then the appearance, disappearance, or the transition from blue to red squares were recorded and tracked throughout the time points. Data from two independent samples were combined to plot the Sankey diagram demonstrating the dynamic change of sprouts and mini-sprouts (*Figure 4I*). The analysis was performed using the Jupyter Notebook graphing library Plotly.

## Immunofluorescence staining of fibronectin and analysis

All reagents for fibronectin immunostaining were injected through the vessel on a chip and kept as follows: 4% (wt/wt) formaldehyde for 1 hr, 0.1% Triton-X for 1 hr, 10% goat serum for 1 hr, fibronectin antibody (Abcam) at 1:50 dilution for 1 day, secondary antibody (anti-rabbit 594 Invitrogen) at 1:100 dilution with 5 µg/ml of Hoechst 33342 for 1 day. 3D volumes at 10 positions from two independent samples of VEGF 100 ng/ml treatment and 2 positions from control (No treatment) were imaged with a ×20 water-immersion objective attached to a Lecia scanning disk confocal microscope. Cells on the parental vessel were identified by GFP expression in the cytoplasm (*Figure 6E*), then segmented (*Figure 6F*) with IMARIS (Bitplane). The averaged fibronectin intensity value of each 3D cell volume and its position were acquired in IMARIS, then each cell was marked as a circle at the corresponding *x* and *y* positions (*Figure 6G*). The averaged fibronectin intensity values and positions of sprouts and mini-sprouts were acquired by identifying cells as surface entities in IMARIS. The averaged fibronectin intensity values of sprouts (following cell) and mini-sprouts were marked as squares at the

corresponding *x* and *y* positions (**Figure 6G**). Ten sprouts, 11 mini-sprouts induced by VEGF 100 ng/ml, and 10 quiescent cells without VEGF treatment were selected to compare fibronectin intensity values (**Figure 6H**). Sprouts often had two compartments of a leading cell and following cells where lumenization occurs. We distinguished the leading sprout and the following sprout for the fibronectin intensity comparison (**Figure 6H**). If a sprout was composed of a single cell with lumen, the cell was classified as the following cell. Seven neighboring cells including a sprout or a mini-sprout in each image were grouped (**Figure 6I**) to analyze the distribution of the ratio of cells having fibronectin levels higher than a threshold value of 30 (dashed line in **Figure 6H**). From 10 images, 21 groups were acquired and used for the analysis (**Figure 6J**).

## Statistical analysis

One-way ANOVA analysis was performed in GraphPad Prism 9.0.0 followed by Tukey's multiple comparisons test. Differences between pairs were concluded to be significant if they had adjusted p values less than 0.05. For all figures, $*p < 0.05$, $**p < 0.01$, $***p < 0.001$, $****p < 0.0001$. If there is no significant difference, it is left blank.

## Acknowledgements

This work was supported by NIH through NCI U54 CA209992 (to AL); Mia Neri Foundation (to T-YK); The Center for Theoretical Biological Physics sponsored by the NSF (Grant PHY-2019745), NSF-PHY-2210291, CPRIT Scholar in Cancer Research sponsored by the Cancer Prevention and Research Institute of Texas (to JNO); a Simons Foundation grant 594598 (to QN). FB was supported by a National Science Foundation grant DMS1763272 (QN).

## Additional information

### Funding

| Funder | Grant reference number | Author |
| --- | --- | --- |
| NIH | NCI U54 CA209992 | Andre Levchenko |
| Mia Neri Foundation | | Tae-Yun Kang |
| NSF | PHY-2019745 | José N Onuchic |
| NSF | NSF-PHY-2210291 | José N Onuchic |
| Cancer Prevention and Research Institute of Texas | CPRIT Scholar in Cancer Research | José N Onuchic |
| Simons Foundation | 594598 | Qing Nie |
| NSF | DMS1763272 | Federico Bocci Qing Nie |

The funders had no role in study design, data collection and interpretation, or the decision to submit the work for publication.

### Author contributions

Tae-Yun Kang, Conceptualization, Resources, Data curation, Formal analysis, Funding acquisition, Validation, Investigation, Visualization, Methodology, Writing – original draft, Project administration, Writing – review and editing; Federico Bocci, Conceptualization, Data curation, Software, Formal analysis, Funding acquisition, Validation, Investigation, Visualization, Methodology, Writing – original draft, Writing – review and editing; Qing Nie, Supervision, Funding acquisition; José N Onuchic, Supervision, Funding acquisition, Writing – review and editing; Andre Levchenko, Conceptualization, Supervision, Writing – original draft, Project administration, Writing – review and editing

### Author ORCIDs

Tae-Yun Kang (iD) http://orcid.org/0000-0002-6122-0870
Federico Bocci (iD) http://orcid.org/0000-0003-4302-9906

Qing Nie ⓘ https://orcid.org/0000-0002-8804-3368
José N Onuchic ⓘ https://orcid.org/0000-0002-9448-0388
Andre Levchenko ⓘ https://orcid.org/0000-0001-6262-1222

Reviewer #1 (Public Review): https://doi.org/10.7554/eLife.89262.3.sa1
Reviewer #2 (Public Review): https://doi.org/10.7554/eLife.89262.3.sa2
Author Response https://doi.org/10.7554/eLife.89262.3.sa3

## Additional files

### Supplementary files
• MDAR checklist

• Source code 1. Source code for mathematical modeling. This includes all codes used in the mathematical modeling for *Figure 2*, *Figure 2—figure supplement 1*, *Figure 3*, *Figure 5*, and *Figure 5—figure supplement 1*.

### Data availability
Source data files for Figures 3, 4, and 6 have been provided and modeling code used for this manuscript is uploaded as Source code 1.

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
