## [Editor Report · eLife assessment]

The authors used an appropriate micro-engineered experimental model of angiogenesis coupled to mathematical model to study the early steps of the angiogenic sprouting. To this end, the authors developed a **convincing** model to predict how VEGF activates Delta-Notch signaling. The work affords **important** new insight into the complex processes involved in the onset of angiogenesis.

---

## [Referee Report · Reviewer #1 (Public Review)]

The authors succeeded in establishing experimental and mathematical models for the formation of new blood vessels. The experimental model relies on temporal imaging of multilcellular projections and lumen formation from a single blood vessel embedded in an engineered extracellular matrix. The mathematical model combines both discrete and continuum elements. It would be helpful to understand how the authors came up with phenotypic classes for analyzing their live imaging data. On the modeling side, it would be useful to see whether the claims about Turing patterns could be supported by either a mean-field model or a more thorough parametric analysis of the discreet continuum model. The authors did a good job in comparing their VEGF/Notch mechanism to the EGF/Notch vulval patterning mechanism in *C. elegans*. The authors might want to look into the literature from studies of the tracheal patterning system in Drosophila when the combined actions of the FGF and Notch signaling specify tip and stalk cells. The similarities are quite striking and are worth noting.

---

## [Referee Report · Reviewer #2 (Public Review)]

Summary:

In this manuscript, the goal of the authors is to understand the process of mature sprout formation from mini-sprouts to develop new blood vessels during angiogenesis. For this, they use their earlier experimental setup of engineered blood vessels in combination with a modified spatio-temporal model for Notch signalling. The authors first study the role of VEGF on Tip (Delta-rich) and Stalk (Notch-rich) patterning. The Tip cells are further examined for their space-time dynamics as Mini-sprouts and mature Sprouts. The Notch signalling model is later supplemented with a phenomenological _random uniform model_ for Sprout selection as a plausible mechanism for Sprout formation from Mini-Sprouts. Finally, the authors look into the role of fibronectin in the Sprout formation process. Overall, the authors propose that VEGF interacts with Notch signalling in blood vessels to generate spatially disordered and co-localized Tip cells. VEGF and fibronectin then provide external cues to dynamically modulate mature Sprout formation from Mini-Sprouts that could control the location and density of developing blood vessels with a process that is consistent with a Turing-like mechanism.

Strengths and Weaknesses

In this manuscript, work motivation, problem definition, experimental procedures, analysis techniques, mathematical methods (including the parameters), and findings are all presented quite clearly. Moreover, the authors carefully indicate whenever they make any assumptions, and do not mix unproven hypothesis with deduced or known facts. The experimental techniques and most of the mathematical methods used in this paper are borrowed from the earlier works of the corresponding authors, and thus are not completely novel. However, the use of these ideas to provide a simple elucidation of the role of VEGF and fibronectin in Sprout formation, in an otherwise complex system, is very interesting and useful. Some of the data analysis methods presented in the paper - (i) quantification of Tip spatial patterns (Fig. 3) and (ii) Sprout temporal dynamics using Sankey diagram (Fig. 4) - seem quite novel to me in the context of Notch signalling literature. Similarly, the authors also provide a new mechanism (VEGF) to obtain disordered Delta-Notch patterning without explicitly including _noise_ in the system (Fig. 2 and Fig. S1). The authors also systematically quantify the statistics of spacing between the Sprouts and show that the Sprouts have a tendency to be away from each other, something that they could also partially recapitulate by additionally including a novel _random uniform model_ for Sprout selection (Fig. 5). Although the association between fibronectin and angiogenesis is known in the literature, in this manuscript, the authors could clearly demonstrate that fibronectin is present in high and low levels, respectively, around Sprouts and Mini-sprouts (Fig. 6). A combination of these findings could then motivate the authors to hypothesize, as mentioned above, a Turing-like mechanism for Sprout formation, something that I find interesting.

Although I find the relative simplicity of the experimental system and theoretical model and the clear findings they generate appealing, some aspects raise a few questions. The authors experimentally find 20 +- 0.08 percent of Tip cells in the model blood-vessels that is consistent with the salt-and-pepper pattern seen in Notch signalling model (~25 %). However, it is not clear to me if the reverse is true, i.e., 25% of Tip cells automatically imply a salt-and-pepper pattern - the authors do not seem to provide a direct experimental evidence. Furthermore, the authors use their Notch signalling model on a regular hexagonal lattice, but there is a large variability in the cell sizes (Fig. 3) in the experimental system. Since it is observed in the literature that signalling depends on the contact area between the neighbouring cells, it is not clear how that would affect the findings presented in this paper. Similarly, since some of the cells are quite small compared to the others, I worry how appropriate it is to express the distance between the Tip cells in terms of _cell numbers_ (Fig. 3). Regarding Sprout classification, as per Table 1, a bridge of two cells is formed as per early-stage-I mechanism for Sprout. On the other hand, the entire data interpretation of experiments seems to be based on early Stage II and matured stage in that same table (also Figs. 3 and 4) in which only one Tip cell seems to be counted per mature Sprout. However, if some Sprouts are formed via early stage-I mechanism, a projection in 2D for analysis would give a count of __two__ adjacent Tip cells, but corresponding to a __single__ Sprout. It could be possible that the presence of such two-cell Sprouts affects the statistics of inter-Sprout distances (Fig. 5). Finally, I find the proposed mechanism of Sprout formation dynamics to be somewhat unsatisfactory. Other than the experimental evidence regarding the spacing of Sprouts and the fibronectin levels around Sprouts and Mini-sprouts (Figs. 4 and 5), there is very little evidence to support the hypothesis about a Turing-like mechanism for Sprouting. Moreover, it seems to me that Turing patterns can appear in a wide variety of settings and could be applied to the current problem in an abstract manner without making any meaningful connections with the system variables. Also, from a modeling point of view, cell migration and mechanics, are expected to take a major part in Sprout formation, while cell division and inclusion would most likely influence Tip-Stalk cell formation. However, it seems that in the present work, these effects are coarse-grained into Notch signalling parameters and the Sprout selection model, thus making any experimental connection quite vague.

Overall Assessment

I feel that the authors, on the whole, do achieve their main goals. Although I have a few concerns that I have raised above, overall, I find the work presented in this manuscript to be a solid addition to the broad field of collective cell dynamics. The authors use well established experimental and mathematical methods while adding a few novel analysis techniques and modeling ideas to provide a compelling, albeit incomplete, picture of Sprout formation during angiogenesis. While the direct application of this work in the context of angiogenesis is obvious, the broad set of ideas and techniques (discussed above) in this work would also be useful to researchers who work on Notch signalling in morphogenesis, collective cell migration, and epithelial-mesenchymal-transition.

---

## [Author Response]

The following is the authors’ response to the original reviews.

**Reviewer #1 (Recommendations For The Authors):**
1. A more thorough analysis of transition boundaries between different types of patterns would further strengthen the conclusions.

We agree that the transition between different patterning regimes should be discussed more quantitatively in the manuscript. Specifically, we identified a highly sensitive parameter range where the disorder in the patterns rapidly increases as a function of the VEGF stimulus. We have improved our discussion of the transition between ‘orderedlike’ patterns and ‘disordered-like’ patterns in the main text as follows: “At relatively low VEGF levels, the patterns were mostly ordered, with small deviations from the expected ‘salt and paper’ geometry with a 25%-75% ratio of TipStalk (Fig. 2D). However, as the VEGF input increased, the fraction of Tips grew and the patterns became sharply more disordered over a relatively narrow range of magnitude of the VEGF input, which could be identified as a highly sensitive area separating more ‘ordered-like’ and ‘disordered-like’ patterns. Finally, increasing VEGF stimuli beyond the highly sensitive area further increased the disorder of the patterns, but with a lower VEGF sensitivity, over several more orders of magnitude of VEGF inputs”.

**Reviewer #2 (Recommendations For The Authors):**
Please refer to the Public Comments above for a broad review. Below, I provide specific concerns that could be addressed.Main comments1. Is the salt-and-pepper model observed for the case when there is no VEGF in the experiments? It would be good to confirm the same. If not, the analysis presented in Fig. 3 could be performed for this case and used as a baseline while referring to the data in Fig. 3.

We thank the referee for the interesting suggestion. The pattern predicted by the model is not strictly salt-and-pepper in absence of VEGF, but the disorder quantified in terms of “incorrect” contacts between Tip cells is considerably lower (see for example the disorder quantification in supplementary figure 1C). We have included the Tip-Tip contact statistics for a case of VEGF=1 ng/ml (100-fold lower that the level used in Fig. 3 compare between model and experiment). In this case, there is clearly more spacing between Tip cells, thus demonstrating how high VEGF stimuli increase the probability of contacts between Tip cells. In the main text, we commented: “As a baseline comparison, the mathematical model with a 100-fold reduction of VEGF stimulus (1 ng/ml) exhibited a Tip-Tip distance statistics more closely comparable with the ‘salt-and-pepper’ model”.

1. The authors mention in the Discussion (end of pg. 7) that...a low level of exogeneous VEGF is essential to induce Delta-NOTCH signalling.. However, in the standard NOTCH signalling (Boareto et al.), we can get the salt-and-pepper pattern without any VEGF. Am I missing something? The authors may want to take a re-look.

1. The authors mention in the Discussion (end of pg. 7) thatHowever, in the standard NOTCH signalling (Boareto et al.), we can get the salt-and-pepper pattern without any VEGF. Am I missing something? The authors may want to take a re-look.

We appreciate the referee’s understanding of the mathematical model. The model used here still exhibits a bistable behavior between the low-Delta and high-Delta cell states even in the absence of VEGF input, as seen for example in the cell state distribution of Fig. 2B, and in agreement with the original model by Boareto et al. This behavior is reflective of the more general applicability of the model, as it describes Delta-NOTCH interactions in various systems. For endothelial cells, VEGF is indeed required to trigger this interaction, but this was not the primary focus of the paper, hence the original model was used. In the text referred to by the reviewer, we are discussing the role,of VEGF based in its known biological effects as well as modeling results. We anticipate that the future further adaptation of the model to,endothelial cells will refine its description of of cell interactions in the absence of VEGF.

1. The size of cells (or spacing between cell nuclei) is highly variable (Fig. 3). Since it is known that the size of cell-cell junctions influences signalling, it would good to at least comment on the same, considering that the model in the paper consists of regular static hexagons. Similarly, it seems desirable to comment on expressing the distance between Tip cells (Fig. 3) in cell length units, when the cell lengths are so variable.

We concur with the suggestion that our consideration of the cell-cell contact size in NOTCH signaling should be clarified in the manuscript.

Sprinzak et al. reported in their 2017 article published in Developmental Cell that the cell-cell contact area does influence NOTCH Signaling. In this article, they found that NOTCH trans-endocytosis (TEC) for pairs with a larger contact width (25µm) is up to five times higher than for pairs with a smaller contact (2.5µm), as observed through the two-cell TEC assay. While TEC correlates with contact width across a range from 1 to 40µm, the values fluctuate significantly in the middle range, particularly when excluding extremely low cell-cell contact areas.

In our experiments, we observed that the cell-cell contact area ranges from essentially infinitesimal corner-to-corner contact to roughly 50µm. We excluded the corner contacts, which might correspond to extremely low cell-cell contact areas, from the Tip-Tip distance measurements as depicted in Fig. 3B. We also made the assumption that variations in cell-cell contact size within tens of microns correlate weakly with the strength of NOTCH signaling. This assumption did not impede our effort to compare the overall trends with results from modeling using hexagonal cells, as shown in Figs 6 D&E. We have included this comment and the corresponding reference to elucidate our assumption in the results as follows: In our experiments, the observed cell-cell contact area varied, spanning from very low (cell corner-to-corner contact) up to approximately 50µm. Previous studies(14, 15) have clearly demonstrated the influence of the cell-cell contact area on NOTCH Signaling, but the values get nosy in the middle range, particularly when excluding extremely low cell-cell contact areas. Reflecting these findings, we excluded the corner contacts, which might correspond to extremely low cell-cell contact areas, from the Tip-Tip distance measurements as depicted in Fig. 3B. We also made an assumption that variations in cell-cell contact size within tens of microns correlate weakly with the strength of NOTCH signaling. This assumption did not impede our effort to compare the overall trends with results from modeling using hexagonal cells, as shown in Figs 3 D&E.

1. The results presented in Fig. 6J are quite striking. However, the number of samples N = 10 and N = 11 seem somewhat low. How does one justify that the findings are not influenced by low number fluctuations?

We acknowledge the reviewer's concerns regarding potential biases stemming from a limited number of samples. The analysis presented in Fig. 6J was specifically designed to complement and support the findings in Fig. 6H. In this context, the counts of sprout and mini-sprout dots correspond to the number of instances "including a sprout" and "including a mini-sprout."

While the counts of sprouts and mini-sprouts in Fig. 6H might seem limited as highlighted by the reviewer, the statistical difference between the two groups was found to be significant. Nevertheless, we expanded our regions of interest to encompass neighboring cells, based on the rationale that the local environment might have closely interacting and similar features. The sample sizes in Figure 6J, represented as N=10 and N=11, equate to an examination of 70 cells and 77 cells, respectively. For instance, in the category "including a sprout," five out of ten groups indicated that all seven neighboring cells in a group exhibited fibronectin levels exceeding a given threshold, translating to 35 cells with fibronectin levels above this threshold. Given that the observed trends in distribution were consistently reasonable across the examinations of both 70 and 77 cells, we would like to state that we are confident in our results.

1. It is written towards the end on pg. 5 that
... although all sprouts indeed formed from mini-sprouts, not all ...
. However, as can be seen from Fig. 4O, Sprouts can also be generated from Stalk cells. This should be corrected.

Thank you for highlighting the discrepancy between our statement on page 5 and the observations in Fig. 4O. While all sprouts undergo a mini-sprout phase, the transition from Stalk to mini-sprout is not always be observed due to the limitations of our observational timeframe. We acknowledge this oversight and adjusted our statement to clarify that sprouts appearing to form directly from Stalks likely passed through an unobserved intermediate mini-sprout stage as follows: We found that all sprouts formed either directly from Stalks or from mini-sprouts, suggesting a non-observed transition from Stalk to mini-sprout due to observational timeframe limitations. Strikingly, however, not all minisprouts persisted and initiated sprout formation.

1. No solid blue bars are shown in Fig. S2A as mentioned in the caption. Kindly correct.

We apologize for the mistake. We have corrected the figure to show the blue bars depicting the experimental measurements for sprout distance probability.

1. How are the high-Delta cells or high-NOTCH cells decided in experiments or simulations? Does it happen that Delta and NOTCH levels are comparable? In that case, what is done? This point could be clarified in the main manuscript or Materials and Methods.

We agree with the reviewer that Tip cell definition should be clarified. In the model, we define a threshold level for cellular Delta to distinguish Tip and Stalk cells, which is now explained in the Methods section “Definition of Tip cells in the model”. As elaborated in the new section, Delta and NOTCH levels are never comparable due to the circuit’s bistable behavior. In experiments, Tip cells based on their key phenotypic characteristic — invasive migration into the surrounding collagen matrix rather than Delta or NOTCH levels. The details can be found in “Precise quantification of Tip cell spatial arrangement suggests disordered patterning in the engineered angiogenesis model” section and Figure 3A.

Minor commentsThere are a good number of typos in the paper. The manuscript should be carefully checked and corrected for the same. Below, I provide a few instances.1. In the abstract towards the end, it should be "understanding" instead of "understating"1. On pg. 5, just before the beginning of the last paragraph, there is a typo "parodied" which should most likely be "provided"1. First paragraph on pg. 6 typo "spouts" instead of "Sprouts"1. Second paragraph on pg. 6, correctly write "testS"1. Near the beginning of pg. 8, should be "*C. elegans*" instead of "C. elegance"1. Figure 1 caption, towards the end, should be "Stalk" instead of "Salk"

We sincerely appreciate your keen attention to detail. we have thoroughly reviewed the manuscript and made the necessary corrections, including those that you have highlighted.

Reviewer #3 (Recommendations For The Authors):Major concern:The authors should discuss in more detail how their work can be used for a better understanding of the angiogenesis process in physiological conditions and in pathological conditions such as post-ischemic revascularization or tumor vascularization.

We have included comments and the corresponding references to clarify the aspect the reviewer suggested: The results in this study can further inform our understanding of angiogenesis in physiological and pathophysiological conditions. In particular, in many circumstances, the levels of VEGF is determined by the degree of hypoxia, which can be highly elevated following oxygen supply interruption, e.g., in wound healing or ischemia, or due to progression of neoplastic growth. Our results suggest that in these cases, formation of sprouts can be dysregulated due to higher incidences of co-localizations of prospective Tip cells. In addition, since these conditions are frequently accompanied by altered synthesis of ECM, the sprout density can increase, which may lead to formation of denser and less developed vascular beds frequently observed as a result of tumor angiogenesis(42, 43). Our results thus suggest that the disorder and higher plasticity of the endothelial cell fate speciation at higher VEGF inputs can be a key contributor to some pathological states associated with persistently hypoxic conditions.